# Fast GCM - ice sheet model coupling software OBLIMAP 2.0, including on-line embeddable mapping routines

**Thomas J. Reerink, Willem Jan van de Berg, and Roderik S.W. van de Wal**

Institute for Marine and Atmospheric research Utrecht, Utrecht University, 3508 TA Utrecht, The Netherlands

*Correspondence to:* Thomas Reerink (tjreerink@gmail.com)

**Abstract.** This paper accompanies the second OBLIMAP open source release. The package is developed to map climate fields between a general circulation model (GCM) and an ice sheet model (ISM) in both directions by using optimal aligned oblique projections, which minimize distortions. The curvature of the surfaces of the GCM and ISM grid differ, both grids may be irregularly spaced and the ratio of the grids is allowed to differ largely. OBLIMAP's stand-alone version is able to map data sets which differ in various aspects on the same ISM grid. Each grid may either coincide with the surface of a sphere, an ellipsoid or a flat plane, while the grid types might differ. Reprojection of e.g. ISM data sets is also facilitated. This is demonstrated by relevant applications concerning the major ice caps. As the stand-alone version applies also for the reverse mapping direction, it can be used as an off-line coupler. Besides, OBLIMAP 2.0 is an embeddable GCM - ISM coupler, suited for high-frequency on-line coupled experiments. A new fast scan method is presented for structured grids as an alternative for the former time consuming grid search strategy, realising a performance gain of several orders of magnitude and enabling the mapping of high resolution data sets with a much larger number of grid nodes. Further a highly flexible masked mapping option is added. The limitation of the fast scan method with respect to unstructured and adaptive grids is discussed together with a possible future parallel MPI implementation.

## 1 Introduction

Ice caps are part of the climate system and interact with the atmosphere and the ocean via various feedback mechanisms. In order to simulate their interaction, ice sheet models (ISMs) need to be coupled with general circulation models (GCMs). In contrast to GCMs which use geographical coordinates, ISMs are often solved on rectangular coordinates, due to the type of the ice dynamic equations. This requires (1) a projection step and (2) a regridding or interpolation step, when coupling an ISM with a GCM.

Studies which achieved a bidirectional coupling of ice sheets with climate models like e.g. Ridley et al. (2005); Mikolajewicz et al. (2007); Vizcaíno et al. (2008, 2010, 2015); Gregory et al. (2012); Lipscomb et al. (2013); Ziemen et al. (2014) barely describe the technical coupling, i.e. the projection and interpolation method they use. If at all, these studies report that a *polar* projection is used. However, the potential impact of grid-area distortion and interpolation accuracy on the mapping justifies an approach in which distortions are minimized in order to optimize the accuracy (Reerink et al., 2010).

Because ISMs are predominantly influenced by the atmospheric forcing, coupling them with an atmospheric GCM (AGCM) is from the ISM perspective self-evident. Nevertheless, if ice shelves are included in the ISM, the coupling with an oceanic GCM (OGCM) makes sense as well because ice shelves are sensitive to the ocean temperatures, which strongly affect the dynamics of the shelves and the ice sheet behind the ice shelve (e.g. Holland and Jenkins, 1999). On the other hand an ISM model provides output in terms of bedrock, surface height and ice sheet distribution which affect the climate and needs to be provided back to the GCM.

Earth system models (ESMs) consist of four basic component models for simulating the atmosphere, ocean, land surface and sea ice. Other components like for instance an atmospheric chemistry model component maybe added to ESMs as well. These components are coupled by ESM component couplers like OASIS3 (Valcke, 2013), MCT (Larson et al., 2005), the ESMF coupler (Hill et al., 2004), the CPL6 coupler (Craig et al., 2005), the CPL7 coupler (Craig et al., 2012) and the C-Coupler (Liu et al., 2014). Valcke et al. (2012,

2016) and Liu et al. (2014) shortly describe and compare these and other couplers. These ESM component couplers have two main functionalities: (1) They serve as a central hub between the different components, i.e. they manage the data transfer and coordinate the execution of the components. (2) The fields are regridded. Nearly all the ESM component couplers use SCRIP (Jones, 1999) for regridding between two spherical coordinate systems.

However, in order to couple GCMs and ISMs, the GCM fields which are defined on a grid representing the curved Earth surface have to be mapped at the ISM grid which coincides with a flat surface, and vice versa. OBLIMAP (Reerink et al., 2010) performs this technical mapping task, which comprises the combination of projection and interpolation in both directions. A consequence of the projection is that even the grid points of a regularly spaced grid end up irregularly spaced, which requires a suited interpolation method. Aside from the additional projection step, the resolution of the GCM and ISM grid often differ largely. Further, mapping ISM fields from a local ISM grid onto a larger scale GCM grid requires a merge of the mapped parts into the existing GCM fields. These and other specific GCM – ISM coupling issues are addressed by OBLIMAP. On the other hand in contrast to ESM component couplers, OBLIMAP is not that much a hub from which more than two ESM components are coupled.

The oblique stereographic (SG) and the oblique Lambert azimuthal equal-area (LAEA) and their inverse projections are suited projections given the constraints involved with this type of geographical mappings, and are therefore available in OBLIMAP. The conformal SG projection preserves angles which is a welcome property for direction dependent geometries and velocity fields. The SG projection is nearly area conservative if the projection is optimally aligned. Areas remain conserved under the LAEA projection, while this projection is nearly conformal if optimally aligned. The area-invariant property of this projection is interesting with respect to conservative mapping. However, note that conserved mapping requires in addition a conserved interpolation method as well. Both projection methods are azimuthal (i.e. perspective), which means that with exception of the point of projection itself, the entire domain can be mapped without any singularities. Either a sphere or an ellipsoid is a natural choice to represent the Earth surface. Therefore in OBLIMAP the SG and LAEA projection can be used both in combination with the sphere and the (default WGS84) ellipsoid.

The interpolation methods have to cope with the fact that the projected grid points end up irregularly spaced with respect to the destination grid points. OBLIMAP uses a quadrant and radius interpolation method (Reerink et al., 2010), which are based on the inverse squared distance weighted interpolation method for irregularly-spaced data (Shepard, 1968). The inverse squared distance weighting function has several practical advantages when interpolating spatial data,

it is suited to identically treat: (1) regular and irregular spaced grid nodes, (2) 1D, 2D and 3D spatial grids, (3) any curved destination surface, i.e. the surface of a sphere, an ellipsoid or a flat plane, (4) a variable number of contributions in the weighting. The latter enables the mapping of data gaps, departure and destination grid domain edges, and masked grid points. This inverse squared distance weighting function is usually combined, mainly for computational performance reasons, with a selection method which excludes remote contributions. As the ratio of the grid resolutions may differ orders of magnitude among OBLIMAP's applications, two selection variants are available. The quadrant interpolation method is used in case a coarse grid is mapped on a fine grid or in case the grids have a similar resolution. It draws a cross through the considered destination grid point and selects in each quadrant the nearest projected contribution. The radius interpolation method is used in case a fine grid is mapped on a coarse grid. It selects the contributions which lay within a certain radius from the considered destination grid point. A reasonable radius typically equals half the departure grid size resolution.

ISMs usually cover a limited local area compared to GCMs. Therefore only a part of the GCM points will participate in the ISM to GCM mapping. These points are distinguished from the remaining points by a participation mask, and will default be merged with the pre-mapped field values of the points which did not participate in the mapping.

Using an optimized projection is fundamental in OBLIMAP's strategy to provide high mapping quality and to obtain results which are close to conservative after to and fro mapping. OBLIMAP uses oblique projections with an optimal standard parallel in order to achieve this. The user only needs to specify the geographical coordinates of the centre of the area of interest in order to obtain this optimal centred projection. With the optimal centred projection the SG projection is close to area preserving. In addition the distortions of the mapped local distances are minimized. The point of projection, which is the only singular point in the mapping, is by definition the counter pole of the center of the area of interest. A diverse set of to and fro mapping tests prove the robustness of OBLIMAP and show accurate results which are close to conservative (Reerink et al., 2010).

Examples of typical input data sets which are used to initialize and force an ISM are topographic data sets including surface height, bedrock level and ice thickness fields, atmospheric forcing data sets containing surface mass balance, surface temperature, refreezing and run-off fields, and data sets containing fields like the geothermal heat flux, the ice surface velocities for validation or for initializing the ice shelf velocity field and for instance the ocean surface temperature which could be used in the ice shelf basal melt parameterization.

For example, in order to equally map their initial topographic and atmospheric and other forcing fields on their preferred ISM grids with an optimal centred projection, Helsen

et al. (2012, 2013, 2016); De Boer et al. (2013, 2014, 2015) used OBLIMAP (Reerink et al., 2010) for preparing their ice sheet modelling input fields. Besides the use of GCM output as atmospheric forcing, the higher resolution fields provided by any regional energy balance model (e.g. RACMO (van Meijgaard et al., 2009) or MAR (Gallée and Schayes, 1994)) might be favoured as forcing of the ISM (Helsen et al., 2013), the more because they deliver direct applicable products like the surface mass balance (SMB) which are of interest for the ice sheet modeller. With the RACMO2.3 data sets for Greenland (Noël et al., 2015) and Antarctica (Van Wessem et al., 2014, 2016) higher resolution atmospheric forcing data sets are available. For instance the RACMO SMB is a field which contains only relevant valid values at ice covered grid points, distinguished by the assignment of a missing value for the remaining grid points. This requires a method which accurately maps masked fields. Therefore OBLIMAP 2.0 provides the new 'masked mapping' method.

At the time of OBLIMAP's first release OBLIMAP typically had to cope with situations in which GCM fields, which are defined on a rather coarse grid ($\sim 1°$), are mapped on ISM grids with a resolution of about 10 km or coarser. In the meantime new high resolution topographic data sets for Greenland (Bamber et al., 2013) and Antarctica (Fretwell et al., 2013) have become publicly available. The large gain in resolution refinement for these data sets confronted OBLIMAP with a performance challenge which has not been an issue at the time of OBLIMAP's first release. Mapping or remapping for instance the entire Greenland area with a 1x1, 2x2 or 5x5 km resolution is much more demanding with respect to the computational time of OBLIMAP's so called scan phase. The scan phase computes the projection of all grid coordinates and calculates the distances over the surface of the destination grid, and based on them the nearby projected points which contribute to the interpolation are selected. With an alternative 'fast scan method' for structured grids OBLIMAP 2.0 realizes a large performance gain and therefore enables the mapping of the high resolution data sets.

In order to simulate the interaction of an ice sheet with the ocean and atmosphere in sufficient detail, high-frequency on-line GCM - ISM coupling is required in which the mutual feedback processes are implicitly included. A solution which is computational efficient, will use embeddable coupling routines and for instance an embeddable ISM. With on-line coupling we mean that the field exchange takes place during a simultaneous GCM - ISM run, which can be achieved with either an external or an embedded coupler, the latter means that the coupling routines are directly called from for instance the GCM code. With OBLIMAP's redesign the mapping routines can be used embedded as well now. The introduction of a so called 'dynamic data object' solves the main I/O bottleneck. This combination makes the OBLIMAP mapping routines suitable for high-frequency on-line coupling, which is one of the main achievements of OBLIMAP 2.0.

In Sect. 3 a few other new key features are described, whereas a complete description is available in the OBLIMAP User Guide (Reerink, 2016). The primary objective of this new OBLIMAP User Guide is to explain how the configuration variables can be configured. Using OBLIMAP's standalone version and applying it on (1) the publicly available topographic and geothermal heat flux data sets and (2) on the RACMO2.3 atmospheric forcing data sets (provided as supplementary material) conveniently generates the input fields at any ISM grid of preference for modelling Greenland and Antarctica. The OBLIMAP-package, including the stand-alone OBLIMAP 2.0 code, two RACMO2.3 data sets, several applications and the OBLIMAP User Guide, is available at the GMD site (see supplementary material http://www.geosci-model-dev.net/???-supplement.zip) and is distributed under the terms of the GNU General Public License.

## 2   General overview of OBLIMAP 2.0

OBLIMAP is divided into two phases: a prior scan phase and a post scan phase. In the scan phase the departure grid points are projected on the destination surface corresponding with the destination grid. For each destination grid point the projected departure grid points which contribute to the interpolation, depending on the selected interpolation method, have to be selected. As this is time consuming, the resulting grid addresses and relative distances are stored by writing them to the scanned indices and distances file: the SID file.

In the post scan phase the dynamic data object (DDO) is initialized by loading the SID file data. Thereafter the fast mapping of multiple fields, layers and records is that fast that the computational time is more or less negligible for most common applications. In a later stage the same mapping with the same departure grid - destination grid combination can be repeated with the scan phase switched off by reading the earlier produced SID file.

Beside the so-called full scan method a fast scan method is available in OBLIMAP 2.0, which is orders of magnitude faster than the full scan method. The fast scan method is applicable to structured grids, like regular Cartesian and gaussian reduced grids. A schematic representation of OBLIMAP 2.0 stand-alone is given in Fig. 1.

In order to couple an ISM on-line with a GCM, one of them should host the other model and the OBLIMAP mapping routines. Due to the complexity of most GCMs, the GCM will be in most cases the obvious host model candidate. The current OBLIMAP 2.0 mapping routines are designed for embedded usage. Of course this approach requires an embeddable ISM as well.

Because in the on-line case the OBLIMAP mapping routines are used embedded, it is recommended to conduct the scanning in both mapping directions off-line and prior to the coupled run. In the initialization phase of the coupled run

both DDOs are loaded by reading both SID files, and the fast mapping can be repeatedly used as shown in the scheme of Fig. 2, where the time steps of the ISM, the GCM and the coupling interval might differ from each other and change over time.

OBLIMAP 2.0 works with a separate configuration file for each mapping direction. A configuration file is an ascii file containing the configuration variables which enables to configure each mapping. The number and order of the listed configuration variables in the configuration file is not prescribed. Those configuration variables which are not listed keep their predefined OBLIMAP settings. The 67 configuration variables are described in the OBLIMAP User Guide (Reerink, 2016). Just one configuration file for both embedded mapping directions will be sufficient if, as recommended in an on-line coupled experiment, the coupled run itself uses only the fast mapping mode.

The post scan configuration parameters can be changed at any time without the obligation of repeating the scan phase. They could even be changed during an on-line coupled experiment. This contrasts with the scan configuration parameters, as soon as a user decides to alter them, the time consuming scan phase has to be repeated.

## 3 New OBLIMAP features and achievements

In this section and in Table 1 the most important new or extended OBLIMAP features and achievements are described.

Before we proceed we emphasize that a distinction is made between data gaps and invalid points in OBLIMAP. If for example a forcing field covers the Earth surface up to a latitude of $87°$South, it means that there will be no departure grid points available within the polar area for the fields which are mapped on an ISM grid for Antarctica. This is what we call a data gap. In such cases with the quadrant interpolation method OBLIMAP succeeds in establishing an reasonable well interpolation of the forcing fields for the polar area on the ISM grid (Reerink et al., 2010). Data gaps are thus areas where no departure grid points are available. By contrast, invalid points are departure grid points which contain an invalid value. With OBLIMAP 2.0 they can optionally be excluded for the interpolation by masked mapping (see Sect. 3.2) in the post scan phase.

### 3.1 Scanning

The objective of the scan phase is to identify all departure grid points which are projected close to each considered destination grid point and thus contribute to their interpolation. Each projected departure grid point which is indeed used for the interpolation of the considered destination grid point is called a 'contribution' of this destination grid point. The number of detected contributing points per destination grid point varies due to the selected interpolation method, but also due to their availability near grid domain borders for instance. In this stage no distinction is made between masked or non masked invalid value contributions, all of them are involved. The masking is a post-scan feature, see Sect. 3.2.

The departure grid indices of each contribution and the distance over the surface associated with the destination grid are written to the SID file. Per destination grid point one line is written to the SID file with the following format (see also the header of each SID file): The indices of the destination grid point are followed by $n$, the number of contributions for this destination grid point, whereafter the $n$ contributions follow with for each contribution its departure grid indices and its relative distance over the destination surface to the destination grid point. $n$ may differ per destination grid point. This format which copes with the fluctuating amount of data per destination grid point is also the basis of the DDO.

In fact OBLIMAP needs in its scan phase the grid coordinates and has to know the grid shape. The actual values at the grid points are not used and therefore not required. The selection of the projected contributions is always based on their relative distance to the destination grid point measured over the destination grid surface. Also this distance calculation requires nothing more than the coordinates of the destination grid point and the projected departure grid point.

Assuming that the ISM grid area is a rectangle defined by the coordinates of the bottom-left and the upper-right corners, the participation mask for the GCM grid can be constructed with negligible computational costs.

#### 3.1.1 Structured and unstructured grids

We distinguish between structured and unstructured grids. The nodes of a 2D structured grid are ordered along the coordinate axes (e.g. by two indices $i,j$) and the neighbour nodes of each node have to be neighbours in the real world space. I.e. based on the grid arrangement of the structured grid the neighbourhood relationships are defined which make them regular connected. With unstructured grids we denote all non-structured grids. Their elements can join in any manner while their neighbourhood relationships require explicit storage to be identified. The Cartesian, rectilinear and curvilinear grid are examples of structured grids, the tessellation and the surface curvature of the grid cell can differ. OBLIMAP supports three surface curvatures: the surface of the sphere, the ellipsoid and the flat plane. Logically rectangular grid and curvilinear grid are used as synonym for the term structured grid elsewhere.

#### 3.1.2 Full scan

In the full scan method for each participating destination grid point a full scan over all projected departure grid points is conducted in order to select all contributions for this destination grid point. Of course this is computational not very efficient, but it is easy to implement and entirely robust with

respect to grid configurations around the geographical poles, cyclic grids and data gaps. The full scan method applies to structured and unstructured grids. OBLIMAP's prior version relied only on the full scan method.

### 3.1.3 Fast scan

The fast scan algorithm uses the fact that the closest projected departure grid point contributions of the previous neighbour destination grid point are known. The closest contribution is called the pivot. Searching the contributions for the destination grid point under consideration can be conducted within a limited block of grid points surrounding the pivot. If this block is constructed sufficiently wide, no relevant contributions will be missed.

At the very first considered destination grid point a full scan over all departure grid points is conducted in order to ensure that the correct contributions are determined. This yields a pivot for the next destination grid point. So for the second considered destination grid point the contribution scan will be conducted within a limited block of departure grid points, whereafter the pivot will be updated for the next destination grid point. This is repeated until the end of the first row of the 2D-grid. When jumping to the next row of the destination grid, the nearest contribution of the neighbour destination grid point at the previous row is taken as the pivot for the first destination grid point at the next row. And so on.

The use of the participation mask in the reverse ISM to GCM mapping direction, complicates this approach. If there is a certain number of non-participating destination grid points at the start of a row, the 'jump approach' will be applied to the first participating destination grid point at this row. There are situations possible in which there is no participating point available at the previous row either. Or no contribution is detected at all at the previous scanned neighbouring points. In such relative rare occasions a full scan has to be conducted for these destination grid points, decreasing the computational time performance.

Evidently in the fast scan method a correct estimate of the block size is essential. A block size radius ($b$) can be either manually specified in the configuration file or automatically estimated by OBLIMAP (default), in the latter case $b$ is estimated locally for each destination grid point. It should be as small as possible to obtain the fastest performance while at the same time it should be large enough to obtain identical results to the full scan method (which serves as the quality-performance bench mark).

The automatic estimate of $b$ by OBLIMAP differs for the quadrant and radius interpolation method. For the quadrant interpolation method $b$ equals the ratio of the local destination grid resolution over the local departure grid resolution. For the radius interpolation method the first term of $b$ equals this same ratio as for the quadrant interpolation method, while an additional second term extends the block size in order to capture the entire area covered by the radius of the radius interpolation method. The second term equals the ratio of this radius over the local departure grid resolution. This yields a sharp minimal estimate of $b$.

Fig. 3 shows the construction of a scan block and the determination of the pivot for a regular example of a GCM to ISM mapping using the quadrant method. Fig. 4 shows the same for a regular example of an ISM to GCM mapping using the radius method. In general the projected grid points are not aligned with the destination grid points, as shown in these examples. Fig. 3 and Fig. 4 describe the main concept of the fast scan method: For each destination grid point a scan over a local block of departure grid points is conducted where the local block has to be well positioned. In fact the fast scan method basically uses the fact that the local block of the previous search will be close to the local block of the next search for structured grids.

The estimate of the local grid resolution is based on the distance to the next grid point of the row. The distance is measured along the surface of the grid. The local grid resolution might vary gradually over the domain of the departure grid and the destination grid. Due to this gradual trend the estimate of the local grid resolution might deviate for neighbour grid points which lay a few nodes away, which affects the estimate of $b$. The results with the fast scan method combined with the automatic estimate of $b$ are identical to the full scan results for the majority of encountered mappings. However, due to the possible under estimation of $b$, distant points could be missed resulting in deviations compared to the full scan result.

A robust automatic fast scan option allows the possibility to avoid the manual configuration of the optimal and correct scan parameter settings. In order to achieve a robust automatic fast scan method, the additional 'dynamic block size method' has been added. With the dynamic block size method the initial local estimated block size is increased step by step, until no additional contributions are detected. The method is coded in such way that only added block borders are checked (for efficiency) and it is consistent with the hereafter mentioned cyclic approach. The dynamic block size method is default switched on, but can be switched off in the configuration file.

In addition, complementary techniques are implemented: 1. In order to deal with cyclic longitudinal domain boundaries. At the east and west longitudinal border of a global departure grid, points at the other side of the departure grid might contain contributions as well. In these cases the block size is extended cyclic by a special treatment. 2. Some equidistant longitude - latitude geographical grids are constructed in such way that the grid row at highest latitude is so close to the geographical pole that all points in this row are situated nearly at the same location but spread in the full longitudinal width over the grid. This is an awkward case for the fast scan method, and OBLIMAP has to carry out a full scan in longitudinal direction in order to obtain identical results to

a full scan. The long-winded description of the exact implementation of these additional techniques is omitted here.

The fast scan method is applicable to any structured grid. Various surface curvatures and grid cell shapes are allowed, the grid nodes are permitted to be irregularly spaced. The distance calculation is supported for distances over the surface of the sphere, over the ellipsoid and in the flat plane. Because the grid neighbourhood relationship of unstructured grids is not self evident from the grid ordering, the fast scan method is not expected to be applicable for unstructured grids. This applies also to regionally refined grids, which show a discontinuous increase in node density at certain internal grid boundaries and therefore do not satisfy the structured grid conditions. In those cases the full scan method has to be used.

A robust automatic fast scan method has been achieved for grids with a gradually varying resolution by including the additional dynamic block size method, yielding identical results to the full scan method.

### 3.2 Masked mapping

Not seldom field variables which have to be mapped are not everywhere available on the departure grid. In such cases it is important that masked points can be excluded for interpolation in order to guarantee the quality performance of the mapping at the borders of the mask. This applies for instance for field variables which are only available at an ice sheet mask, like the surface mass balance (SMB) and refreezing fields as produced by RACMO. In another example one might argue that the missing data at the exterior of the bottom topography field, might be a minor issue in a single mapping because of the remoteness of the problem to the area of interest. However, after several times of to and fro mapping the error will propagate into the area of interest.

With OBLIMAP's masked mapping, invalid masked points are ignored for the interpolation. If for a certain destination grid point the nearest projected departure grid point has an invalid mask, default this destination grid point will be set to that invalid value. However, this behaviour can be changed for each mapped field individually by altering the mask criterion, as with the other option all valid contributions are considered irrespective if the nearest contribution has an invalid mask. For each mapped field a separate invalid value can be specified. The mask of each field for the masked mapping is shaped by the pattern of the invalid value for that field. Each mask is allowed to vary in time, and might vary per vertical layer in case the masking is based on a spatial 3D field. Default a mask of a certain field can be based on the invalid value pattern of that field itself, but the mask can also be based on the pattern of one of the other mapped fields. Because masked mapping is a post-scanning option it can be switched on and off at any time.

Excluding the masked area might allow a significant decrease of $\alpha$, the angle which controls the standard parallel, which induces a further optimalization of the projection.

In case of a masked mapping and a raised message level OBLIMAP will inform the user about an optimal masked $\alpha$, estimated by:

$$\alpha = \arcsin\left(\frac{1}{R}\sqrt{\frac{1}{2\pi}\left(\mathrm{COUNT(mask)}\right)\Delta x \Delta y}\right) \quad (1)$$

with $R$ the Earth radius, $\Delta x$ and $\Delta y$ the ISM grid spacing in the $x$ and $y$ direction and the 2D mask is 0 for invalid values and 1 otherwise. Eq. (1) equals Eq. (2.2) in Reerink et al. (2010) except that the total number of ISM nodes $N_x N_y$ is replaced by the number of ISM points which have a valid value, being $\mathrm{COUNT(mask)}$.

### 3.3 Embedded mapping

Once a certain grid combination has been scanned the scanning phase can be omitted if this mapping is repeated off-line or on-line afterwards. With OBLIMAP 2.0 the data of the SID file (for each destination grid point the indices of each contribution and its relative distance to the destination grid point) is stored in the dynamic data object (DDO). With that the off-line performance becomes faster for the multiple field, multiple layer and multiple record mapping. But more importantly, it solves the I/O bottleneck for high-frequency on-line coupling.

The redesign and recoding of the OBLIMAP package enables the embedded calling of the same mapping routines which are used in the off-line stand-alone mode. Embedding the OBLIMAP mapping routines basically requires five code additions to the host model: 1. Adding the OBLIMAP initialization routine, which reads the OBLIMAP configuration variables, in the initialization of the host model. 2. Loading the DDOs for both mapping directions by reading both the SID files in the initialization of the host model. 3. Declaring a vector of spatial 3D fields where the vector length equals the number of mapped fields. 4. Calling the OBLIMAP mapping routines in both directions within the host model time loop. 5. Deallocate the DDOs in the finalizing stage of the host model. OBLIMAP's Application Programmer Interface (API) for additions 2 to 5 is outlined in Fig. 5. The first addition concerns a call to the *initialize_config_variables()* routine, which is usually called one level higher.

The introduction of the DDO in combination with OBLIMAP's redesign solved the I/O bottleneck for high-frequency coupling and enables the embedded calling of the mapping routines, which makes OBLIMAP suitable as GCM - ISM coupling software for on-line coupling projects.

### 3.4 Nearest point assignment

The 'nearest point assignment' is a post scan alternative to the quadrant and radius interpolation method. Instead of interpolating the nearby projected source points on the destination nodes, with this option each destination node obtains the field value of the nearest projected source point disregard-

ing any other contributions. This method can be combined with masked mapping, in that case a destination node will be always invalid if the nearest projected point has an invalid mask. Regardless which interpolation method has been used during the scan phase, the 'nearest point assignment' can be used in the post scan phase. This option can be considered in case both grids have about the same resolution. The 'nearest point assignment' performs faster than both interpolations methods in the post scan phase.

### 3.5 Vincenty method for distances on the ellipsoid

An ISM to GCM mapping projects ISM grid points on the Earth ellipsoid. In that case the interpolation requires the distance over the curved surface for the Shepard distance weighting between each projected ISM point and the GCM point. The geodesic, which is the shortest route between two points on the Earth's surface along the great circle, can locally be approximated accurately by the geodesic on the auxiliary Earth sphere. OBLIMAP uses this estimate default for the distances on the Earth ellipsoid, because this saves computational time in the scan phase and because the contributions are located close to each other compared to the Earth radius so the deviations will be small. However, OBLIMAP 2.0 provides the option to calculate the precise geodesics for the ellipsoid by Vincenty's method. Vincenty's inverse numerical approximation (Vincenty, 1975a) is implemented in OBLIMAP.

### 3.6 Mapping multiple layers of spatial 3D fields

In the most common OBLIMAP applications spatial 2D geographical fields are mapped for none, one or more time records. With OBLIMAP 2.0 it is possible to map spatial 1D, 2D and 3D fields for none, one or more time records. The mapping of 3D fields is in fact limited to the mapping of several parallel vertical layers where each layer simply uses the same mapping, a so called 2D + 1D approach (c.f. Liu et al., 2014). The layers are assumed to be close to each other in comparison with the Earth radius, because for each layer the same Earth radius is applied. Furthermore the horizontal grid distribution of all layers is assumed to be identical to the grid distribution of the top layer. And the fields are only interpolated in the horizontal directions for each vertical layer.

The new OBLIMAP 2.0 netcdf I/O routines automatically detect the spatial dimension of each input field and whether it contains the (unlimited) time dimension. The spatial 2D and 3D fields, including or excluding the time dimension, can be mapped simultaneously and in arbitrary order. With this combination a convenient way of mapping dimensionally different fields is achieved.

### 3.7 Automated selection of scan parameters

The radius method is recommended as soon as the destination grid resolution is four times larger than the departure grid resolution (Reerink et al., 2010). In all other cases the quadrant method is favoured. Based on this criterion the interpolation method is default automatically selected in OBLIMAP 2.0 by checking the ratio of the grid resolutions of the grid centres. However it is also possible to select the interpolation method manually.

Similarly the optimal radius is default automatically determined for the radius method, if the radius method is selected. OBLIMAP checks whether the geographical grid has cyclic longitudinal grid borders, if so default the cyclic mode will be switched on automatically. Finally OBLIMAP can also determine automatically the optimal angle $\alpha$ which controls the standard parallel of the projection, but this is not default the case.

### 3.8 Data architecture, messaging and the User Guide

OBLIMAP 2.0 stores all fields in a vector of fields with each field a spatial 3D field. In case the spatial dimension of a certain field is lower than 3D, its dimension is reduced when written to a netcdf. This applies also to symmetric dimensions, they will be default reduced if they are fully symmetric in one dimension. The field vector is reused and updated for successive records or time steps.

The geographical scientific climate database conventions are followed in the fully recoded I/O-netcdf interface, which has become highly flexible and which has been largely automized. The package has been professionalized in the sense that for any user warning or error we endeavour to provide a meaningful message (see items 11-13 in Table 1). The embedding of the OBLIMAP mapping routines requires the addition of a minimum of code in the host model, for that reason programmer error messages are added in order to prevent improper software usage. The new OBLIMAP User Guide (Reerink, 2016) serves the user to correctly configure the mappings and OBLIMAP's options.

## 4 Performance and applications

### 4.1 Computational time performance

Sixty benchmark mapping experiments have been used, representing a diverse set of mappings which differ in number of nodes, grid resolution, mapping direction, interpolation method, location and thus also in projection. This relative arbitrary set of benchmarks has been used to evaluate the computational efficiency of the fast and full scan method.

The mapping experiments in Fig. 6 are subdivided by different symbol colours: ISM to GCM mappings are plotted red and purple for the quadrant and radius interpolation method respectively. GCM to ISM mappings are plotted blue and light blue for the quadrant and radius interpolation method respectively. The typical error in the time measurements is twice the size of the plot symbol in the figure, and is obtained by repeating the mapping many times.

The part of the Earth's surface which is covered by ice is limited in comparison with the entire ocean - atmosphere surface. Therefore the number of grid nodes which are involved in the mapping is significantly lower than the total number of GCM nodes. This is relevant with respect to the time performance plotting. Therefore the performance is plotted in Fig. 6 as function of $N$ the number of participating destination grid points multiplied with the number of departure grid points. The data is plotted on a logarithmic scale for both axes in Fig. 6, which improves the visualization of the trends compared to the otherwise rather sparse clustered plotting.

The time spent in the full and fast scan routines are measured by including built-in Fortran time counting routines inside the OBLIMAP code, and are shown for this set of mappings respectively in Fig. 6a and Fig. 6b. Fig. 6c shows the gain factor (equal to the full scan time divided by the fast scan time) for these mappings if the fast scan method is used instead of the full scan method. Fig. 6d shows the total gain achieved per individual 2D field in case the post scan fast mapping is used instead of the full scan method. For the latter the fast mapping time is divided by the number of fields, layers and records because they might differ for each of the mappings.

Fig. 6a shows a strong increase of the computational time with $N$ for the full scan method. Clearly visible are the separate branches for the different mapping directions though their trend is similar. In contrast no significant differences are caused due to the selected interpolation method. As can be seen from Fig. 6b the computational time for the fast scan method is much lower, and though it increases with $N$, it is important to observe that it levels off for higher $N$ which is also reflected in Fig. 6c where the gain factor strongly increases with $N$. The scattering in Fig. 6b is larger than in Fig. 6a because the size of the local scan block is sensitive to the individual grid configurations. Because the fast scan times are relative fast, this causes a relative large influence. As a consequence the gain factor and the total gain factor are influenced by this scatter of the fast scan method measurements. The large values of the gain factor in Fig. 6c show that the fast scan method is orders of magnitude faster than the full scan method. Fig. 6d shows a strong increasing trend of the total gain factor up to $\sim 10^6$ at $N = 10^{12}$. For example the fast mapping time of one individual 2D field is about $\sim 10^{-2}$ seconds for $N = 10^{10}$, which corresponds with an high resolution application in which the Greenland area is mapped between a $0.1°$ GCM grid and a 5x5 km ISM grid.

## 4.2 Masked and non-masked mapping applications

This section shows masked mapping for a relevant set of applications and different masking issues are discussed. At the same time this section demonstrates that OBLIMAP's standalone version is a powerful tool which is able to map diverse kinds of topographic and forcing data sets onto any ISM grid configuration with an optimal oblique projection.

The publicly available high resolution topographic data sets are remapped (reprojected from a polar to an optimal oblique aligned projection for the ellipsoid) for each area on a certain ISM grid of preference, i.e. with the desired grid extensions and grid resolution. The atmospheric forcing data sets which are defined on a reduced gaussian grid of the regional RACMO2.3 model, are mapped from the sphere to the same ISM grids. The geothermal heat flux field which is defined on a global regular longitude-latitude grid, is also mapped from the sphere to these same ISM grids. Besides, these different data sets cover a wide resolution range and map the two major ice sheets Greenland and Antarctica, in addition the Antarctic Peninsula example shows how a local subregion is mapped with its own optimal oblique projection.

Table 2 lists the mapping parameters for each mapping on the three ISM grids in the various mapping examples. In case a data set is remapped, the coordinates are projected twice, however the fields are only interpolated at the final destination grid in order to minimize the mapping error. The first projection leaves the field data unaffected, only the (x,y)-coordinates are converted to (longitude, latitude)-coordinates. This task can be conducted by the oblimap convert program which is part of the OBLIMAP-package (see the OBLIMAP User Guide (Reerink, 2016)).

### 4.2.1 Plotting projected data

Some general remarks are made concerning the plotting of the pre and post mapped data displayed in Figs. 7-12. The high quality data itself is saved in netcdf files. In order to visualize those fields, pythons matplotlib and its basemap extension are used to script the plotting. For plotting fields which are defined on grids which are based on geographical coordinates, a projection has to be specified with basemap.

The plotting interpolates the fields once projected by the plotting. Though the selection of the plotting projection can be independently and arbitrary chosen from the mapping projection, we used for most GCM field plots a plotting projection which is similar to the mapping projection. The fields on the ISM grid are plotted as true grid values, i.e. no plotting interpolation has been applied. In several subfigures one black or a few coloured contours are plotted on top of the data. In several subfigures the ETOPO data set (Amante and Eakins) is used as background for the masked mapped areas or for a surrounding area with a constant value. Usually this concerns the ocean basin and remote areas. Plotting the data on the ETOPO data set also serves as a check because all coastline contours should coincide.

### 4.2.2 Topographic fields for Greenland

The publicly available topographic data set for Greenland (Bamber et al., 2013) contains the surface topography, the bedrock topography and the ice thickness for the present day situation and is projected by (Bamber et al., 2013) with a po-

lar SG projection on an ISM grid with a 1x1 km resolution. In a post-processing phase some manual corrections are applied to this 1x1 km ISM grid (Bamber et al., 2013).

In order to obtain an optimal centred projection this data set has been remapped by first applying the inverse polar projection on all coordinates of this data set (see Fig. 7a and Fig. 7c) and thereafter this result is mapped with an optimal centred oblique SG projection on an ISM grid with a 5x5 km resolution (see Fig. 7c and Fig. 7d).

No masked mapping is used for the mapping of the surface topography on the ISM grid in Fig. 7b because it concerns a field which literally levels off to sea level (the zero level). Because the bedrock topography contains missing values (see the white bottom corner areas in Fig. 7c), a masked mapping is used for this field. Resulting in a properly mapped mask border, as can be seen at the bottom right corner of Fig. 7d.

Note that in fact it would be possible to directly map the irregular spaced measured data points, but then the manual applied corrections are lacking.

### 4.2.3 Atmospheric forcing fields for Greenland

The present day RACMO2.3 atmospheric forcing data set for Greenland (Noël et al., 2015) which is provided as supplementary material and contains the surface mass balance (SMB), the surface air temperature, the surface refreezing, run-off and other fields, is defined on a reduced gaussian grid with an approximate horizontal resolution of about 11 km (see Fig. 8a and Fig. 8c). These fields are mapped with the same projection on a grid with the same 5x5 km resolution and extent as used in Sect. 4.2.2. The SMB field in Fig. 8a only contains valid values for ice covered grid points, and is therefore masked mapped on the ISM grid (see Fig. 8b). The same applies for the refreezing in Fig. 8c which is also masked mapped on the ISM grid (see Fig. 8d).

### 4.2.4 Topographic fields for Antarctica

The publicly available Bedmap2 topographic data set for Antarctica (Fretwell et al., 2013) contains the surface topography, the bedrock topography and the ice thickness for the present day situation and is projected by (Fretwell et al., 2013) with a polar SG projection on an ISM grid with a 1x1 km resolution.

In order to obtain a data set which can be mapped on any (local) grid with OBLIMAP, the inverse polar projection is applied on all coordinates of this data set (see Fig. 9a and Fig. 9c). Thereafter this result is mapped with the same polar SG projection on an ISM grid with a 20x20 km resolution (see Fig. 9c and Fig. 9d). Though this might seem superfluous, the advantage is that from this longitude - latitude based data set a grid of any grid extent and resolution can be created with OBLIMAP. In addition any optimum centred local grid can be created from this data set as well, like for instance the local area of the Antarctic Peninsula (see Sect. 4.2.6). We

choose the common polar SG projection for entire Antarctica, however given the position of the Antarctic continent a slightly oblique projection might in fact yield the optimal projection.

No masked mapping is used for the mapping of the surface topography on the ISM grid in Fig. 9b because it concerns a field which literally levels off to sea level (the zero level). Because the bedrock topography contains missing values (see the white corner areas in Fig. 9c), a masked mapping is used for this field. Resulting in a properly mapped mask border, as can be seen at the corners of Fig. 9d.

### 4.2.5 Atmospheric forcing fields for Antarctica

The present day RACMO2.3 atmospheric forcing data set for Antarctica (Van Wessem et al., 2014) which is provided as supplementary material and contains the SMB, the surface air temperature, the surface refreezing, run-off and other fields, is defined on a reduced gaussian grid with an approximate horizontal resolution of about 27 km (see Fig. 10a and Fig. 10c). These fields are mapped with the same projection on a grid with the same 20x20 km resolution and extent as used in Sect. 4.2.4. The SMB field in Fig. 10a only contains valid values for ice covered grid points, and is therefore masked mapped on the ISM grid (see Fig. 10b). The same applies for the refreezing field in Fig. 10c which is also masked mapped on the ISM grid (see Fig. 10d)

The refreezing field in Fig. 10c is an example of a source field with an inadequately defined value for the missing data which equals zero in this case. This zero value is inconvenient here because the field values themselves reach zero in the interior of Antarctica. Coincidently this implies that a masked mapping based on an invalid value which is taken equal to zero, will affect the mapping of the zero contour inside the interior of Antarctica as well. Despite this error the masked mapping is still to be preferred over the non masked mapping in this case because the latter one yields large errors at grid points in the vicinity of the coastline. Here we circumvent this problem by using the ice-cover mask, which is co-distributed as part of the RACMO2.3 data set. In this case the ice-cover mask is used as the mask for the masked mapping of the refreezing, which illustrates the flexibility of the masked mapping options. Preferably the invalid value for the missing values of the source fields have a value outside the range of the actual field values in order to avoid this problem.

### 4.2.6 Local mapping of the Antarctic Peninsula

In contrast to the polar projection used in Sects. (4.2.4-4.2.5), a local mapping of the Antarctic Peninsula demonstrates an oblique projection example with the same data sets. The Bedmap2 surface topography (Fig. 9a) and the Bedmap2 bedrock topography (Fig. 9c) have been mapped on a local 5x5 km ISM grid for the Antarctic Peninsula with an optimal

centred oblique projection without using a mask (see Fig. 11a and Fig. 11b). The RACMO2.3 SMB (Fig. 10a) and the RACMO2.3 refreezing (Fig. 10c) have been masked mapped on the same local 5x5 km ISM grid with the same optimal centred oblique projection (see Fig. 11a and Fig. 11b).

#### 4.2.7 Mapping the geothermal heat flux

The spatial variable geothermal heat flux (Shapiro and Ritzwoller, 2004) which is defined on a global regular $1° \times 1°$ longitude - latitude grid (see Fig. 12a) for the present day situation, is another forcing data set which can be mapped on the same grid. In Fig. 12b this geothermal heat flux has been mapped on the same 5x5 km ISM grid with the same optimal centred projection for Greenland as in Sect. 4.2.2. In Fig. 12c the geothermal heat flux has been mapped on the same 20x20 km ISM grid with the same optimal centred projection for Antarctica as in Sect. 4.2.4. Finally in Fig. 12d the geothermal heat flux has been mapped on the same 5x5 km ISM grid with the same optimal centred projection for the Antarctic Peninsula as in Sect. 4.2.6.

### 4.3 Coupling and embedding

An experiment in which a GCM is on-line coupled with an ISM consists of much more than the technical coupling task. Perfectly mapped fields may require a successive downscaling step. Several decisions have to be made concerning issues like to which degree the on-line coupling will be conducted, which fields are available for a particular GCM - ISM combination and can be effectively used, which model time steps and coupling time step will be used and does that require certain time averaging prior to each coupling step, and for which fields only the perturbations will be used in the coupling. Examining the results should learn whether the coupling is numerically stable, if the feedback mechanisms do properly work and to which extent the resolution differences limit the coupling of the models. Presenting coupled results requires the evaluation of these issues and the description of the used GCM and ISM, but that is far beyond the scope of this paper.

Instead we only shortly report that we coupled the IMAU-ICE model with CLIMBER-2 (Petoukhov et al., 2000) by using OBLIMAP's mapping routines embedded. We benefit from CLIMBER's Fortran77 implementation in which all relevant variables for the coupling are globally defined. This makes it possible to embed CLIMBER in the ice sheet model without major recoding of CLIMBER. Therefore in this case the ice sheet model is taken as the host model and the OBLIMAP mapping routines are embedded in the ice sheet model as well. The low CLIMBER resolution certainly limits the coupling degree, but appeared to be suitable for practical and technical learning purposes, because the IMAU-ICE - CLIMBER coupled model technically eas-

ily operates on a laptop. In general however, due to its complexity, it is preferable to take the GCM as the host model.

## 5 Discussion

OBLIMAP 's full scan method is robust and suitable for any GCM - ISM grid combination regardless of the irregular spacing of the grids and their arbitrary ordering including unstructured grids. The full scan method is used for benchmarking, but can also be used in special cases like embedding a flow line model or embedding a low resolution model like CLIMBER. The drawback of the full scan method is its slow performance. This becomes a serious bottleneck for larger grid combinations, i.e $N \gg 10^{12}$. The fast scan method is orders of magnitude faster, especially for larger grid combinations. The fast scan method is applicable to structured grids. It would require an additional index mapping of the external stored neighbourhood relationships, to enable the fast scan method for unstructured grids. This index mapping methods are expected to vary across different unstructured grid applications. OBLIMAP's fast scan method reduces the search time. The same objective has been addressed in OASIS4 (Redler et al., 2010) in a parallel approach, though Valcke (2013) reports that its development is not pursued.

There are three variants of the fast scan method available in OBLIMAP 2.0: 1. A fixed block size radius ($b$) is manually specified, each search uses the same $b$ which should be large enough. 2. A local $b$ is estimated by OBLIMAP itself. 3. The second variant is extended with the dynamic block size method. The differences in computational time performance between the variants is limited. There is a trade off between the best performance and robustness. The third variant is robust in any situation, and is therefore the default. Robust means that the results are identical to those obtained by the full scan method. OBLIMAP 2.0 contains an option which determines the (fast) scan parameters automatically, in order to avoid that expert knowledge is required to configure the scan parameters. This option is default switched on.

Some simple longitude - latitude grids include a full row of longitude nodes near or at $90°$ North. In that case the location of $N_{\text{LON}}$ nodes coincide, while in practice the field values of these nodes are not identical. In order to guarantee the robustness of the automatic fast scan in these cases, a full longitude scan is applied in the vicinity of the polar area for all nodes with $|\text{latitude}| \geq 87°$ (as discussed in Sect. 3.1.3). The scanning of this kind of grids themselves is usually fast enough, so this is not a major issue. But this full longitude scan at high latitudes also applies to other high resolution grids, leading to a performance decrease. Automatic detection, whether a situation requires a full longitude scan, would be an advantage in a future OBLIMAP release, avoiding unnecessary performance loss.

If the quadrant interpolation method is used in combination with the full scan method, it is possible that in one of the quadrants a relative remote contribution is detected compared to the other quadrants. This can occur due to nearby missing data while the quadrant interpolation method continues searching for the nearest contribution until it is found. The remote contribution has a very limited influence on the interpolation due to the distance weighting. However, with the fast scan method the limited block size deselects this remote by-catch and accordingly the results are not identical but deviate insignificantly. In fact the fast scan result is favourable in this case.

OBLIMAP is able to map between models which differ largely in resolution. In particular if the destination grid is much coarser than the departure grid, the computational time increases inevitably for both the fast scan method and the fast mapping due to the large radius in the radius interpolation method. For large data sets it is important to note that if a sudden tremendous slow down is encountered while $N$ is increased step by step, it is likely that the size of the processor memory is the bottleneck. In that case it is recommended to switch to a platform with a larger processor memory.

We conclude that the default automatic fast scan method is robust, indispensable for large grid combinations (i.e. for large $N$) and can be safely used by non-expert users.

Various examples for different resolution combinations show that the masked mapping works well, i.e. artefacts are absent in the vicinity of the mask borders and the fields represent realistic values as shown in Figs. 7-11. The masked mapping method is indispensable for products like the SMB, the refreezing and the run-off, because their values differ strongly along the mask border. In case fields like the surface topography are mapped frequently to and fro, masked mapping prevents the propagation of artefacts into the domain of interest. Besides, these figures show the high quality of the masked and non-masked fields with high resolution.

OBLIMAP is a powerful tool to map data sets which might differ in grid surface curvature, grid type, grid resolution and grid extent on an equal arbitrary ISM grid by the same optimal centred projection. Although 'optimal' and 'centred' are the recommended preferences, neither of them is a prerequisite. This is demonstrated for the topographic, atmospheric forcing and geothermal heat flux data sets by the applications in Sect. 4. Similar, the ice surface velocity data set for Greenland (Rignot and Mouginot, 2012) could also be mapped on an ISM grid for Greenland. The topographic and geothermal heat flux data sets for Greenland and Antarctic are public available. The present day time-averaged RACMO2.3 atmospheric forcing data sets for Greenland (Noël et al., 2015) and Antarctic (Van Wessem et al., 2014) are provided as supplementary material.

The redesign and recoding of OBLIMAP in combination with the DDO introduction ensures that the embeddable OBLIMAP 2.0 mapping routines are suited to bridge the technical task of on-line coupling in both directions. As indicated in Sect. 3.3 and in Fig. 5, five code items have to be added in the host model in order to embed the OBLIMAP mapping routines. OBLIMAP is subdivided in the standard components: 'Initialize', 'Run' (map and inverse map) and 'Finalize', this allows direct embedding in the ISM, in the GCM or in an ESM component coupler. Note that the mapping routines pass on all fields as an argument, which makes the embedment of OBLIMAP low intrusive. The embedment strategy might depend on the specific GCM – ISM combination as well on the coupling approach: one way or two way on-line coupling. In case a two way on-line coupling is considered, we suggest to embed the ISM in the GCM or in the ESM component coupler. The ISM has then to be recoded in 'Initialize', 'Run' and 'Finalize' components, but this approach avoids invasive modifications of the GCM code. For this reason we plan to separate the initialization phase and the time loop for IMAU-ICE. At the same time, an embeddable ISM allows the simultaneous embedment of multiple ISM domains like e.g. Greenland and Antarctica, each with its own projection and configuration file. In addition it enables the simultaneous simulation of several ice cap domains by the ISM in case they are mutual connected by the sea level evolution, similar with the approach of De Boer et al. (2014).

Like the C-Coupler (Liu et al., 2014) the 3D field mapping concerns a 2D + 1D mapping, in the sense that the horizontal mapping includes the 2D interpolation. Each vertical layer is treated with the same 2D horizontal interpolation but is not interpolated in the vertical direction by OBLIMAP. Returning the vertical layers just as vertical records is a conscious choice, because it keeps the best flexibility. For example it allows the vertical coordinate to change without affecting the mapping, i.e. avoiding a repeated scan phase. This is particular important regarding the vertical zeta coordinate in ISM models which usually not only changes in time but even changes per grid node in time. In this way the vertical grid is allowed to match with either a real or scaled coordinate and could differ per field, again without affecting the mapping. It allows direct downscaling if one wishes, which in that case saves one interpolation step. This is all possible without losing much on the performance, because the vertical interpolation is computational straightforward and at low cost.

The projection step is an essential obligatory step in case two models run on differently curved surfaces. This is the case for GCM – ISM coupling when a GCM which runs on the surface of the Earth Sphere is coupled with an ISM which runs on a flat plane. In contrast, regridding between two ESM components which both run on the same Earth Sphere surface requires only the interpolation step and the projection is not needed as one stays on the same curved surface. The additional (inverse) projection step in GCM – ISM coupling has a few important consequences for the cross grid search. Due to the projection, it is in general a priori unknown how the grid nodes of the two grids are related to each other, the projected nodes can end up anywhere depending on the

projection. The scan method has to robustly cope with that. Other specific requirements in GCM – ISM coupling are: (1) The ISM grid concerns a local part of the GCM which requires a neat treatment of this mapped ISM domain border. (2) Mapping ISM fields from a local ISM grid onto a larger scale GCM grid requires a merge of the mapped parts into the existing GCM fields. (3) The range of resolution ratios is much larger, i.e. often the ISM grid resolution is much finer than that of the GCM. These specific requirements are the cause that GCM – ISM coupling is not standard included in the existing ESM component couplers like OASIS3 or OASIS3-MCT, the ESMF, CPL6 and CPL7 couplers, or the C-Coupler.

OBLIMAP addresses these specific GCM – ISM coupling issues, whereas the ESM component couplers are complex hubs from which a variable amount of ESM components are coupled. Nevertheless, there are also many functional similarities like cross grid searching, interpolating, off-line generation of weight factors, reading the weights and using them for a fast interpolation, generic field exchange, embedding and strategies to parallelize the high cost cross grid search method. OBLIMAP's masking facility is comprehensive and highly flexible compared to the other couplers, it is independent of the scan phase. Each field can be masked based on a user specified masking value, this masking pattern is allowed to change in time and per vertical layer. Masking of a certain field can also be based on the mask pattern of another field. OBLIMAP actually does not store the weight factors like SCRIP does, instead the indices of the contributions and their distances are stored in the SID file. This offers the flexibility in a post scan phase to change the mask and the distance weighting exponent. The SID file and DDO have been designed such that the required processor memory is minimized. Non-participating destination points are not stored in the SID file. OBLIMAP does not use the matrix multiplication like several ESM component couplers. If a fine grid is mapped on a coarse grid the large number of contributions per destination grid point cause a rather large amount of non-zero diagonals in the sparse matrix. Instead OBLIMAP uses the direct access to the indices and distance of the contributions via the DDO (which has to be loaded only at initialization), which allows a very fast evaluation.

OBLIMAP does not include an area conserved interpolation method. Jones (1999) shows that first order area conservative interpolation is much less accurate (especially for fields with large gradients) than e.g. bilinear interpolation. Jones (1999) therefore presents a more accurate second order area conservative interpolation. The second order variant needs the gradient of the field, which is problematic because this does not allow prior off-line generation of the interpolation weights, and is field dependent. Therefore the ESM component couplers use the first order area conservative remapping of SCRIP (Jones, 1999) which is able to regrid between two spherical coordinate systems.

For large scale ice caps it is important that flow directions are not affected by the projection in order to stay close to the physical representation of the models. This means that a SG projection is used which slightly deforms the area of each cell. The combination of a projection with an area conservative mapping leads to large errors: If for instance the area of a cell shrinks by 1%, the value of that cell will increase by 1% to compensate due to the area conservation. However, the area mismatch is compensated after the reverse mapping. OBLIMAP's strategy is to reduce the area distortion by using oblique projections and an optimal standard parallel. The accuracy of the direction dependent ice flow physics is preferred over exact area conservation, the latter would be only possible in combination with a LAEA projection. The conservation of the GCM – ISM coupling should be judged by comparing the results after to and fro mapping. This requires adequate tests, like those carried out by Reerink et al. (2010). The quadrant and radius interpolation method which are based on the inverse squared distance weighting show results close to conservation. OBLIMAP uses the radius method to obtain a representative estimate for mapping from fine to coarse resolution grids.

OBLIMAP is dedicated to the GCM - ISM mapping and coupling task in both mapping directions and developed from that perspective. However, OBLIMAP might be very well applicable to other geophysical mapping problems because the included mapping components like the projection, the regridding and the fast scan method for structured grids are based on a generic implementation.

Could OBLIMAP become an ESM component coupler? Actually it is not our goal as there are several ESM component couplers available. Adding to OBLIMAP the sphere to sphere mapping without a projection step, will be straightforward. In addition a MCT (Larson et al., 2005) combination with OBLIMAP could then provide the hub functionality. An inventory of the design of the ESM component couplers seem to show that this matches well with OBLIMAP. Concerning the primary functionalities there seems thus to be no obstruction.

Adaptive grids require repeated scanning each time one of the grids has changed. With the current serial OBLIMAP code only lower-frequency on-line coupling of low resolution adaptive grid models is feasible using the fast scan method. Embedding the mapping routines in combination with allowing repeated scanning, changes the list of code addition as indicated in Sect. 3.3 as follows: The second item is replaced by an allocation statement inside the time loop in the host model, with a successive call to the embeddable scan routine for each mapping direction. The fifth item, the deallocation of the DDO needs to move inside the time loop at its end. However, a parallel implementation will be beneficial for on-line coupling of adaptive grids and will extend the possibilities.

It is possible to implement an efficient scalable parallel domain decomposition of the full and fast scan method. The

results of this parallel MPI implementation are expected to be bitwise identical for a changing number of processors. The same is expected for the fast mapping scheme. The challenge will be to reduce the used processor memory per node, as in a straightforward parallel approach each core will allocate its own copy, which will limit the scalability for large grid applications. A parallel full scan method will serve applications which use unstructured grids. A parallel fast scan method will serve applications which use structured grids, in particular if this concerns on-line coupled adaptive grid applications which require the scan phase each time one of the grids change.

An extra performance gain could be realized when the single initial full scan over the departure grid points at the start of a destination grid row (which is at some points required), is split up over more processors. However, this concerns a more complex parallel implementation. Apart from this another strategy for adaptive grids could be to make use of the pivots of the previous mapping step by remembering and updating them. This works under the assumption that the adaptive grid changes are locally smooth. It could potentially realize a large performance gain for a parallel and serial approach.

## 6 Conclusions

With a significant larger range of applications than its prior version, OBLIMAP 2.0 has become much faster and easier to configure. The OBLIMAP User Guide has been added to precisely describe OBLIMAP's comprehensive options. The increase in performance, the ability to map grids with much more grid nodes, the extensive and flexible way of masked mapping, and the fact that the OBLIMAP mapping routines can be used embedded in an high-frequency on-line coupled application, are among the main achievements of OBLIMAP 2.0 while OBLIMAP's high accuracy and robustness is maintained.

The power of OBLIMAP is its ability to map various data sets, which are defined on different curved surfaces and may largely differ in grid resolution, extent and type, by an optimal centred projection on one destination grid of arbitrary configuration. This potential has been demonstrated by relevant examples in which topographic, atmospheric forcing and geothermal heat flux fields from various data sets have been mapped on grids for the two major ice caps.

The fast and fully serial OBLIMAP 2.0 package is lightweight and suitable to run on a laptop. Its stand-alone version can be installed and compiled within a couple of minutes on any platform. A future parallel approach, using MPI, offers the possibility of an additional performance gain in a next OBLIMAP release.

## Code and data availability

The OBLIMAP 2.0 code, the present day averaged RACMO2.3 atmospheric forcing data sets for Greenland and Antarctica and the OBLIMAP User Guide are available as supplementary material (http://www.geosci-model-dev.net/???-supplement.zip) and are distributed under the terms of the GNU General Public License. The source code of OBLIMAP can be downloaded through *svn checkout https://svn.science.uu.nl/repos/project.oblimap* from the OBLIMAP svn repository or by a git checkout from OBLIMAP's Github: https://github.com/oblimap/oblimap-2.0. If any problems are encountered with the code, please feel free to contact us (tjreerink@gmail.com).

*Acknowledgements.* We thank Melchior van Wessem for providing the RACMO2.3 data of Antarctica. We thank Bas de Boer, Michiel Helsen, Heiko Goelzer and Sarah Bradley for their user feedback over the years on the OBLIMAP package. This project has been funded by Kennis voor Klimaat (KvK) and by a grant from the Netherlands Earth System Science Centre (NESSC) from the Netherlands Organization for Scientifical research (NWO). The RACMO2.3 data sets are provided within a NESSC-WP3 collaboration. We thank F. Colleoni, an anonymous reviewer and J.G. Fyke for their comments and suggestions which improved the manuscript.

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

**Table 1.** Shortlist of the new OBLIMAP 2.0 features and changements. The stars indicate the categories of the new features. For the full list and more details see the supplementary OBLIMAP User Guide. The terms in bold are used for short reference to the described feature.

| Feature number | Most important | Extended functionality | User comfort | Description of the new OBLIMAP 2.0 features |
|---|---|---|---|---|
| 1 | ★ | ★ | | The mapping of large grid combinations has become feasible with the **fast scanning** option, as a result of the dramatic performance improvement in the scanning phase. |
| 2 | ★ | ★ | | With **masked mapping** points which are part of an invalid mask will not contribute to the interpolation. The invalid mask covers the area for which the field values of a certain user specified field matches with a certain user specified invalid value. |
| 3 | ★ | ★ | | With **multiple masked mapping** each field can use its own mask. This mask is allowed to vary in time, and might vary per vertical layer in case the masking is based on a spatial 3D field. |
| 4 | | ★ | ★ | The redesign of OBLIMAP allows **embedded calling** of the OBLIMAP mapping routines. |
| 5 | ★ | ★ | | The introduction of a **dynamic data object** avoids superfluous reading of the scanned file. This improved the post scanning phase performance, in particular in combination with multiple record, multiple field and multiple layer (3D) mapping. |
| 6 | ★ | ★ | | **On-line coupling** of an ISM with a GCM is possible by using the OBLIMAP routines embedded. Prior off-line scanning is recommended for both mapping directions, so the embedded OBLIMAP mapping routines can use the fast post scan mapping relying on the dynamic data object. |
| 7 | | ★ | | **Nearest point assignment**, a post scan alternative to the quadrant and radius interpolation method. I.e. each destination node obtains the field value of the nearest projected point, which implies that no interpolation is required. This option can be considered in case both grids have about the same resolution. |
| 8 | ★ | ★ | | **Multiple field , multiple record** and **multiple layer mapping**. The 2D and 3D fields can be mapped simultaneously, while each field might also contain the unlimited time dimension. In the embedded case the fields will be mapped at each coupling time step. |
| 9 | | ★ | | The precise calculation with the **Vincenty method** of the great distance over the ellipsoidal arc is added as an option. |
| 10 | | ★ | ★ | With the **automatic scan option** the scan parameters are determined by OBLIMAP itself. |
| 11 | | | ★ | Automatic OBLIMAP advice concerning the correct and optimal settings of the scan parameters. |
| 12 | | | ★ | Extended OBLIMAP messaging including four levels of message intensity. |
| 13 | | | ★ | Automatic dimension shape determination while reading the netcdf input files. |
| 14 | | | ★ | Separate configuration files have to be used for each mapping direction. |
| 15 | | ★ | | The ISM grid is allowed to be irregularly spaced. |
| 16 | ★ | | ★ | An OBLIMAP User Guide accompanies OBLIMAP 2.0 |

**Table 2.** The grid sizes $N_x$, $N_y$ and the grid resolution $\Delta$ of the three ISM grids which result from the mapping applications in Sect. 4.2, using the oblique SG projection for the sphere or the WGS84 ellipsoid with the projection parameters $\alpha$ and the coordinates of the projection centre ($\lambda_M$,$\phi_M$).

| ISM grid | $N_x$ | $N_y$ | $\Delta$ (km) | $\alpha$ (°) | $\lambda_M$ (°) | $\phi_M$ (°) |
|---|---|---|---|---|---|---|
| Greenland | 301 | 551 | 5 | 7.1 | 319.0 | 72.0 |
| Antarctica | 281 | 281 | 20 | 19.0 | 0.0 | -90.0 |
| Peninsula | 271 | 351 | 5 | 5.54 | 293.5 | -70.2 |

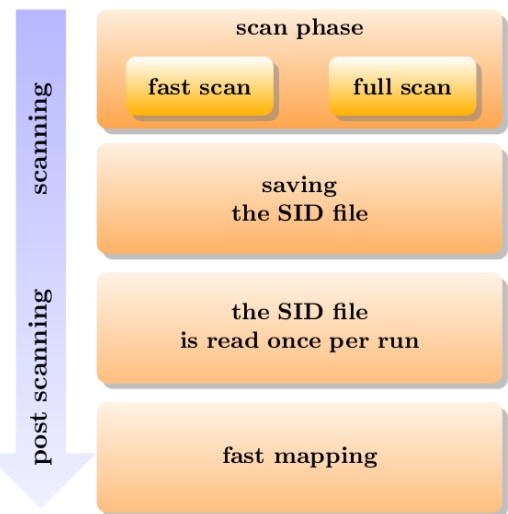

**Figure 1.** Schematic representation of the OBLIMAP 2.0 stand-alone version. The scanning consists of the scan phase and results in a SID file. The post scanning consists of reading the SID file and loading its content in the DDO, whereafter the fast mapping of multiple fields, layers and records can be repeated as often as required.

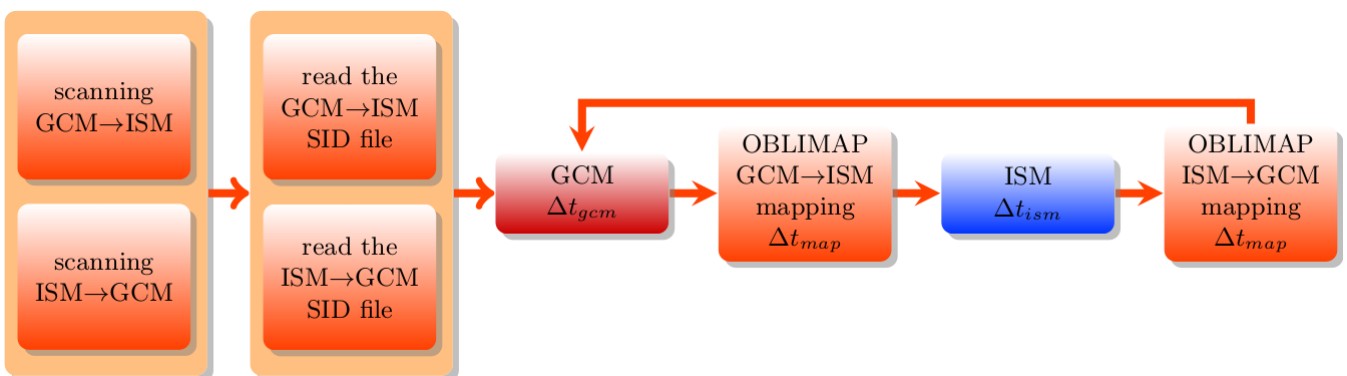

**Figure 2.** Schematic representation of an embedded ISM within a GCM run. At the initialization of the on-line coupled run the SID files are read, which have been created by off-line scanning prior to this run. The GCM and ISM are coupled with the embedded OBLIMAP routines at each coupling interval $\Delta t_{map}$. The GCM and the ISM evolve with their own time step $\Delta t_{gcm}$ and $\Delta t_{im}$ depending on their specific numerical stability criteria.

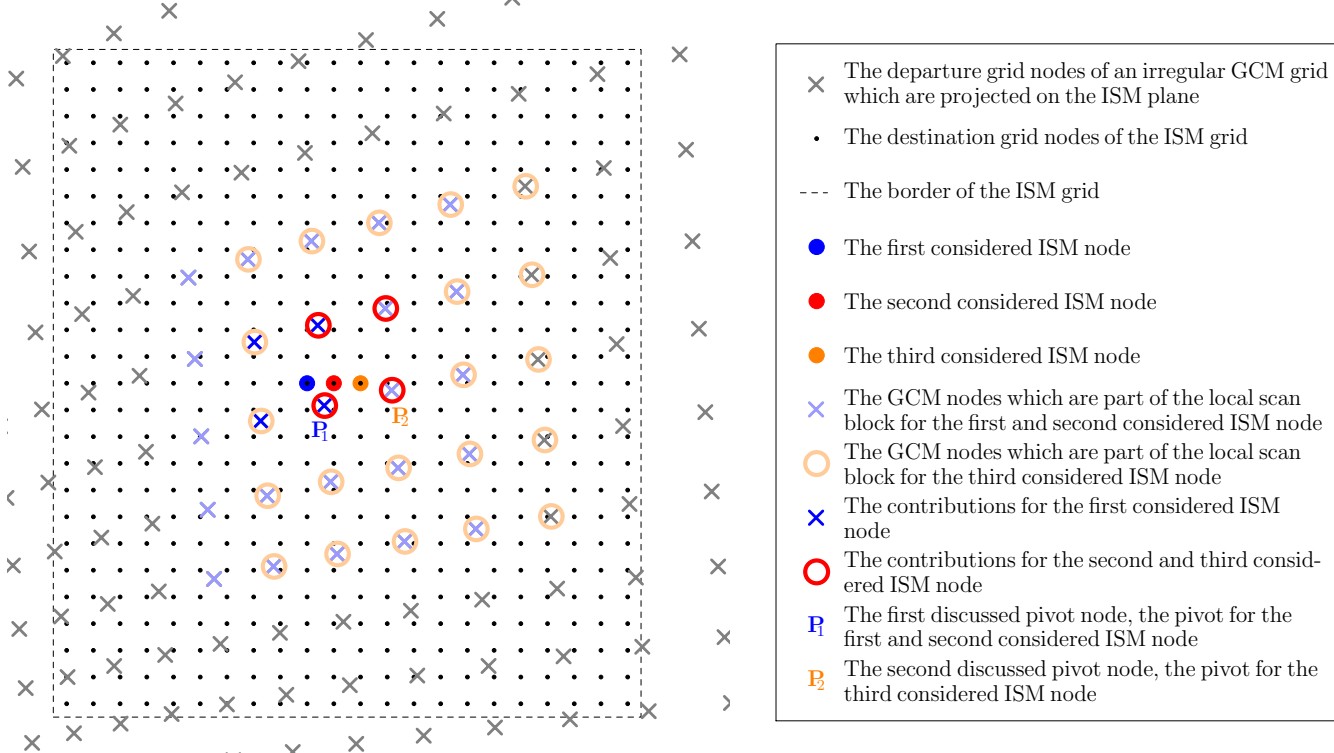

**Figure 3.** Situation sketch of the procedure to find a next scan block for a GCM to ISM mapping, supposed that the quadrant interpolation method is used. Consider the situation for which the four contributions (✕) are found for the first considered ISM node (●). In order to find the contributions for the second considered ISM node (●), a scan over a local block is conducted. This local block is constructed by taking the nearest contribution (the pivot) and extending the block in all directions by $b$. In this case $\mathbf{P}_1$ is the pivot and the block is extended by two ✕ nodes in each direction. For the second considered ISM node this results in a different set of contributions (◯). Because $\mathbf{P}_1$ is the nearest contribution for the second considered ISM node as well, $\mathbf{P}_1$ stays the pivot. Therefore the scan block remains unchanged in the next scan. Though for the thrid considered ISM node (●) the contributions are the same as for the second considered ISM node, now $\mathbf{P}_2$ becomes the pivot of the thrid considered ISM node. So the next scan is conducted over the ◯-marked GCM nodes.

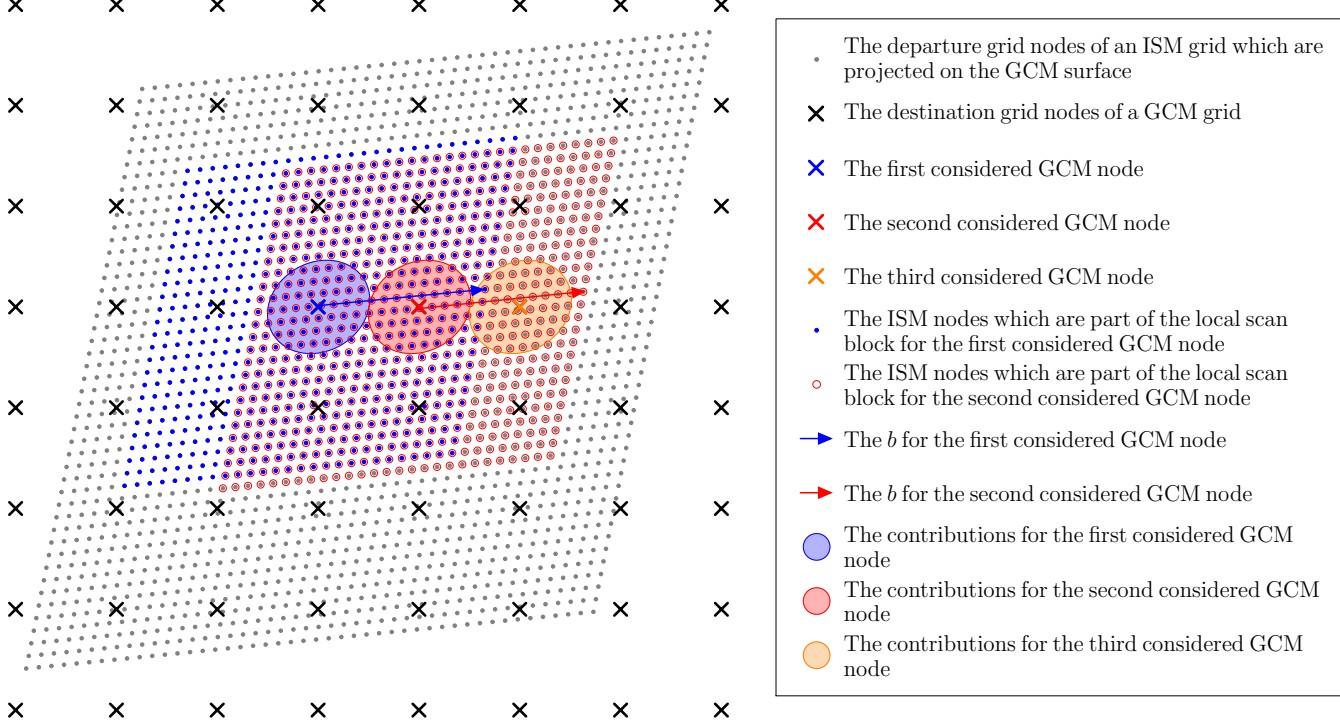

**Figure 4.** Situation sketch of the procedure to find a next scan block for an ISM to GCM mapping, supposed that the radius interpolation method is used. Consider the situation for which the contributions (the ISM nodes in the blue shaded circle) are found for the first considered GCM node (✕). In order to find the contributions for the second considered GCM node (✕), a scan over a local block is conducted. This local block is constructed by taking the nearest contribution (the pivot) and extending the block in all directions by $b$ (indicated by the arrows in the figure). In this case the pivot is the ISM node under the ✕-marker and the •-marked block is extended as indicated by the blue arrow. The contributions for the second considered GCM node lay in the red shaded circle. The pivot shifts to the GCM node under the ✕-marker. The scan for the thrird considered GCM node (✕) is conducted over the ○-marked ISM nodes in order to find the orange shaded contributions. Thereafter the pivot shifts again to the ISM node under the ✕-marker for the next scan.

```fortran
MODULE example_gcm_host_model_module

CONTAINS
  SUBROUTINE example_gcm_host_model()
    ! A BLOCK OF: USE statements of the Host Model
    USE oblimap_configuration_module ,     ONLY: dp , C
    USE oblimap_mapping_module ,           ONLY: oblimap_ddo_type , oblimap_deallocate_ddo
    USE oblimap_embedded_mapping_module , ONLY: oblimap_initialize_embedded_mapping , &
      oblimap_embedded_gcm_to_im_mapping , oblimap_embedded_im_to_gcm_mapping
    IMPLICIT NONE

    ! A BLOCK OF:   Declaration statements of the Host Model
    REAL(dp) , DIMENSION(C%number_of_mapped_fields ,C%NX   , C%NY   ,C%number_of_vertical_layers) :: ism_field
    REAL(dp) , DIMENSION(C%number_of_mapped_fields ,C%NLON , C%NLAT ,C%number_of_vertical_layers) :: gcm_field
    REAL(dp) , DIMENSION(C%number_of_mapped_fields ,C%NLON , C%NLAT ,C%number_of_vertical_layers) :: prev_gcm_field
    TYPE(oblimap_ddo_type)                                                                         :: ddo_gcm_to_im
    TYPE(oblimap_ddo_type)                                                                         :: ddo_im_to_gcm

    ! Output: −
    CALL initialize_ISM()

    ! Output: ddo_gcm_to_im , ddo_im_to_gcm
    CALL oblimap_initialize_embedded_mapping(ddo_gcm_to_im , ddo_im_to_gcm)

    ! A BLOCK WITH: The initialization of the Host Model

    ! Start time loop of the Host Model:
      ! A BLOCK WITH: The Host Model time loop code (including the update of gcm_field)

      ! Keeping the previous gcm_field: 1. For merging with points which do not participate in the mapping.
      !                                 2. Eventually for time interpolation.
      prev_gcm_field = gcm_field

      ! Output: ism_field
      CALL oblimap_embedded_gcm_to_im_mapping(ddo_gcm_to_im , gcm_field , ism_field)

      ! In/Output: ism_field
      CALL embedded_ISM(time_start_ISM , time_stop_ISM , ism_field)

      ! Output: gcm_field
      CALL oblimap_embedded_im_to_gcm_mapping(ddo_im_to_gcm , ism_field , prev_gcm_field , gcm_field)

      ! A BLOCK WITH: The Host Model time loop code
    ! End time loop of the Host Model:

    ! A BLOCK WITH: The finalization of the Host Model

    ! Output: −
    CALL oblimap_deallocate_ddo(ddo_gcm_to_im)
    ! Output: −
    CALL oblimap_deallocate_ddo(ddo_im_to_gcm)
  END SUBROUTINE example_gcm_host_model

END MODULE example_gcm_host_model_module
```

**Figure 5.** A schematic outline shows how to use the OBLIMAP API for embedding an ISM with OBLIMAP in a GCM host model. The *initialize_ISM* and *embedded_ISM* are hypothetic ISM routines which are not part of OBLIMAP but embed the ISM.

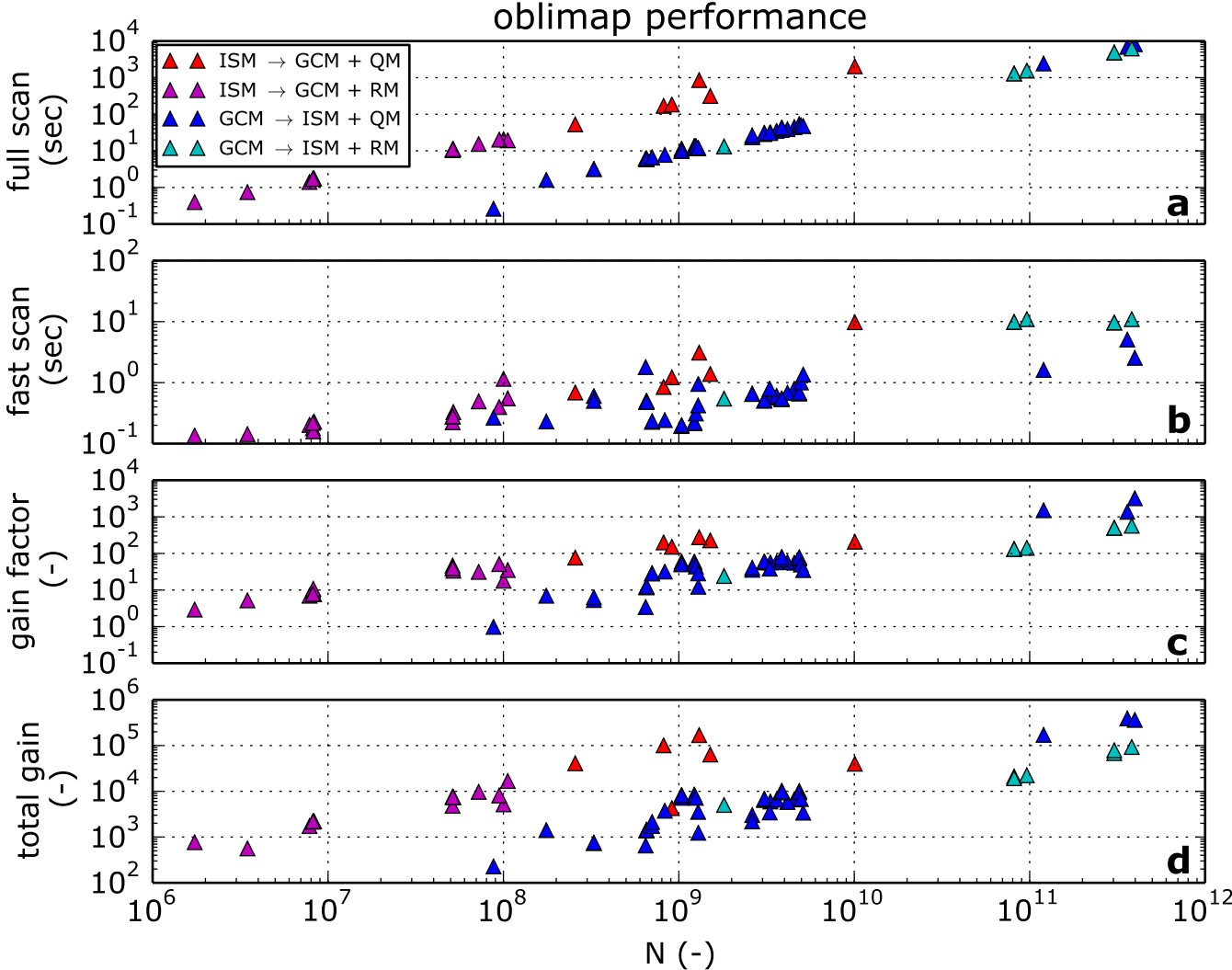

**Figure 6.** Sixty mapping experiments representing a diverse set of mappings which differ in number of nodes, grid resolution, mapping direction, interpolation method, location and thus also in projection, have been used to compare the time performance of the full scan method (**a**) and the fast scan method (**b**) as function of $N$, the number of participating destination grid points multiplied with the number of departure grid points. Subfigures **c** and **d** show the gain factor if respectively the fast scan method or the fast mapping per individual 2D field is used instead of the full scan method. Four colours distinguish between experiments which differ in mapping direction and which use either the quadrant interpolation method (QM) or the radius interpolation method (RM).

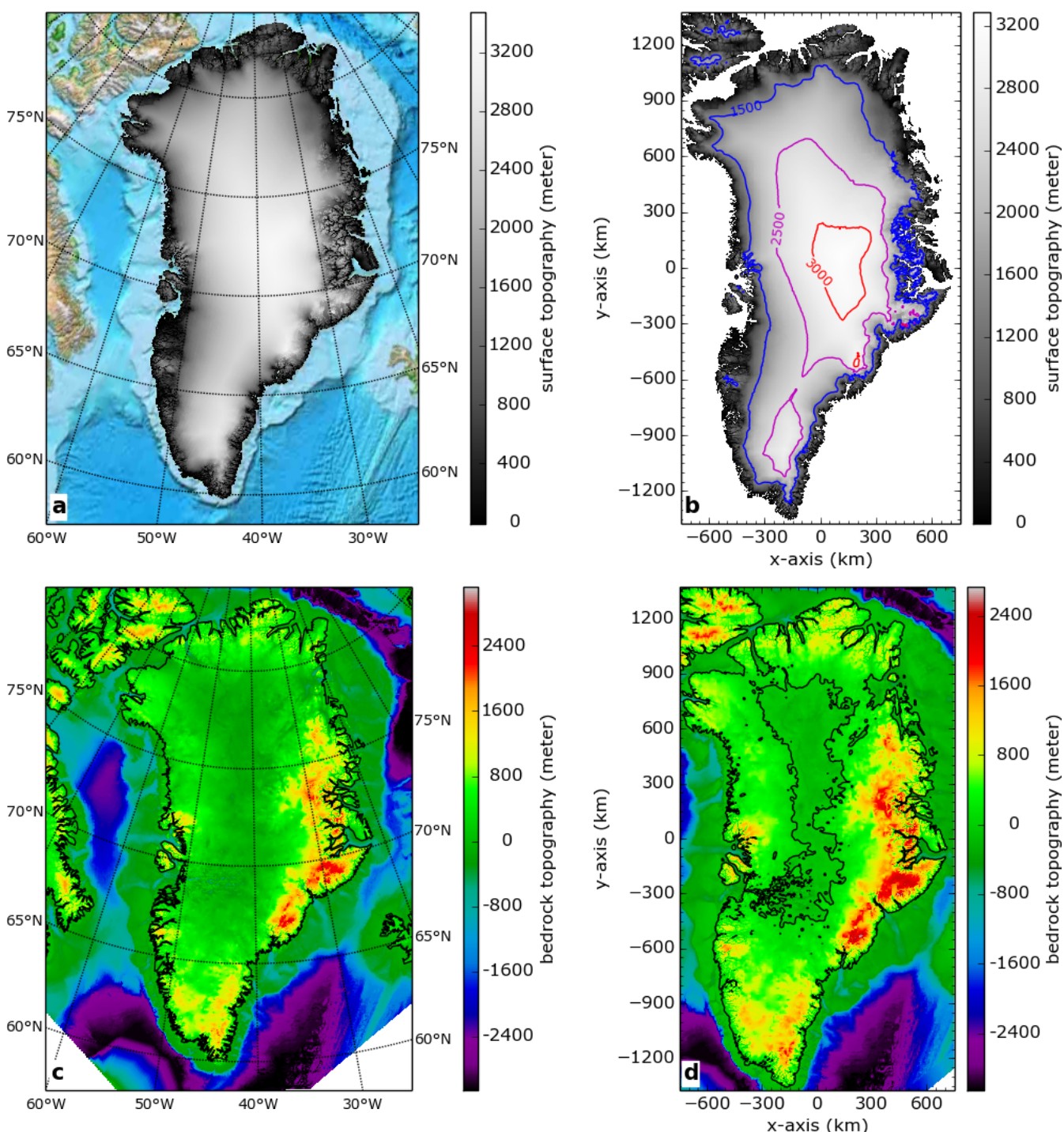

**Figure 7.** Subfigure **a** and **c** show respectively the surface topography and the bedrock topography of the 1x1 km resolution data set for Greenland after an inverse projection on the Earth WGS84 ellipsoid. In **a** a coloured ETOPO background replaces the part at sea level. In **c** the missing values are white coloured and the continental contour is plotted black. In **b** the surface topography has been mapped on a 5x5 km ISM grid with an optimal centred projection without using a mask, but points at sea level have been plotted white in order to visualize the coastline contours. In **d** the bedrock topography has been masked mapped on a 5x5 km ISM grid with an optimal centred projection, the bottom right corner shows the proper resulting mask border. In **b** a few coloured contours are plotted on top of the data, and in **d** the zero contour is plotted black.

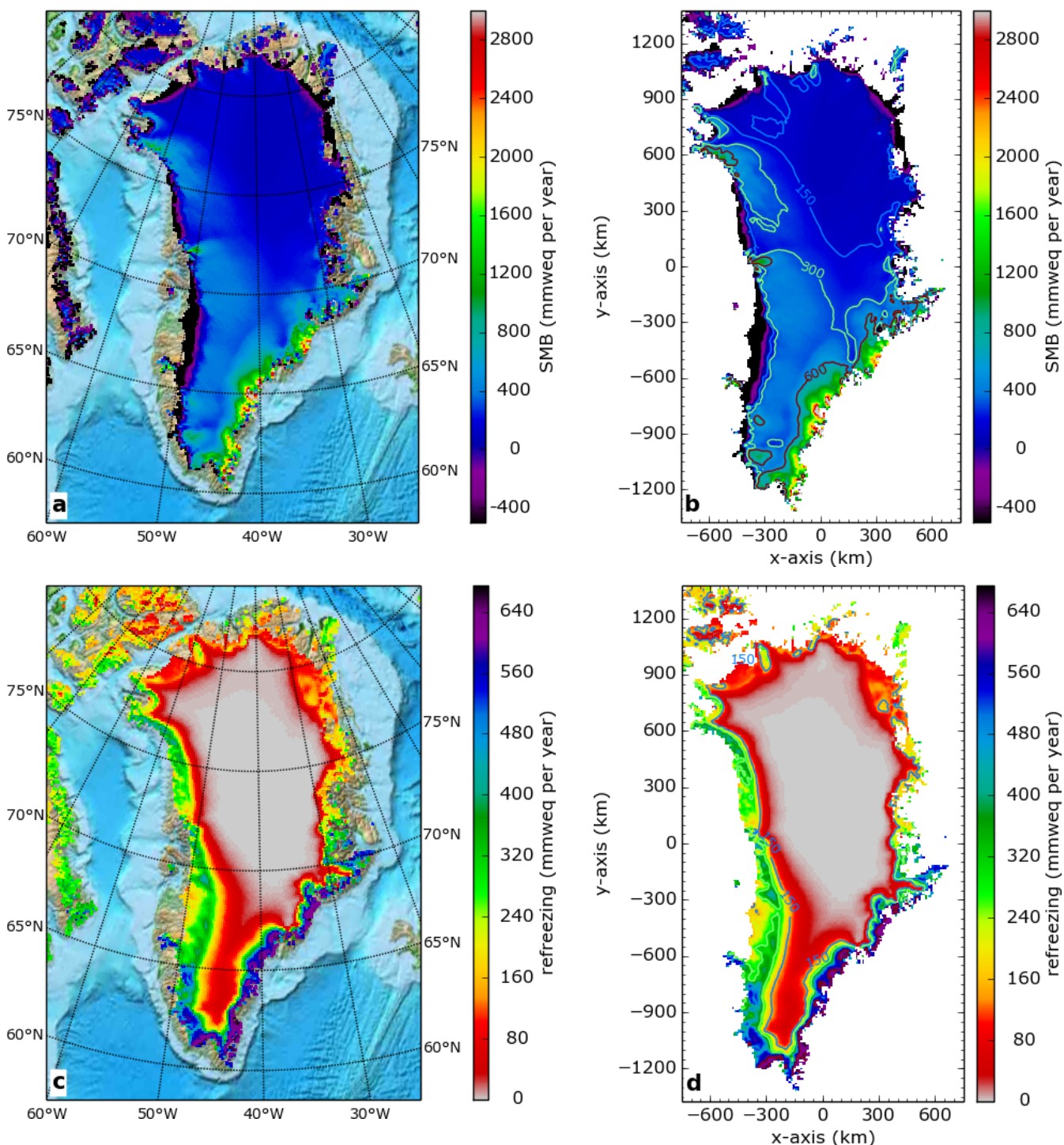

**Figure 8.** Subfigure **a** and **c** show respectively the surface mass balance (SMB) and the refreezing of the 1958–1989 time average of the RACMO2.3 data set for Greenland on a reduced gaussian grid with an horizontal resolution of about 11 km, the coloured ETOPO background replaces the masked area. In **b** and **d** the SMB and the refreezing have been masked mapped on a 5x5 km ISM grid with an optimal centred projection. Both fields are expressed in millimeter water equivalent (mmweq per year).

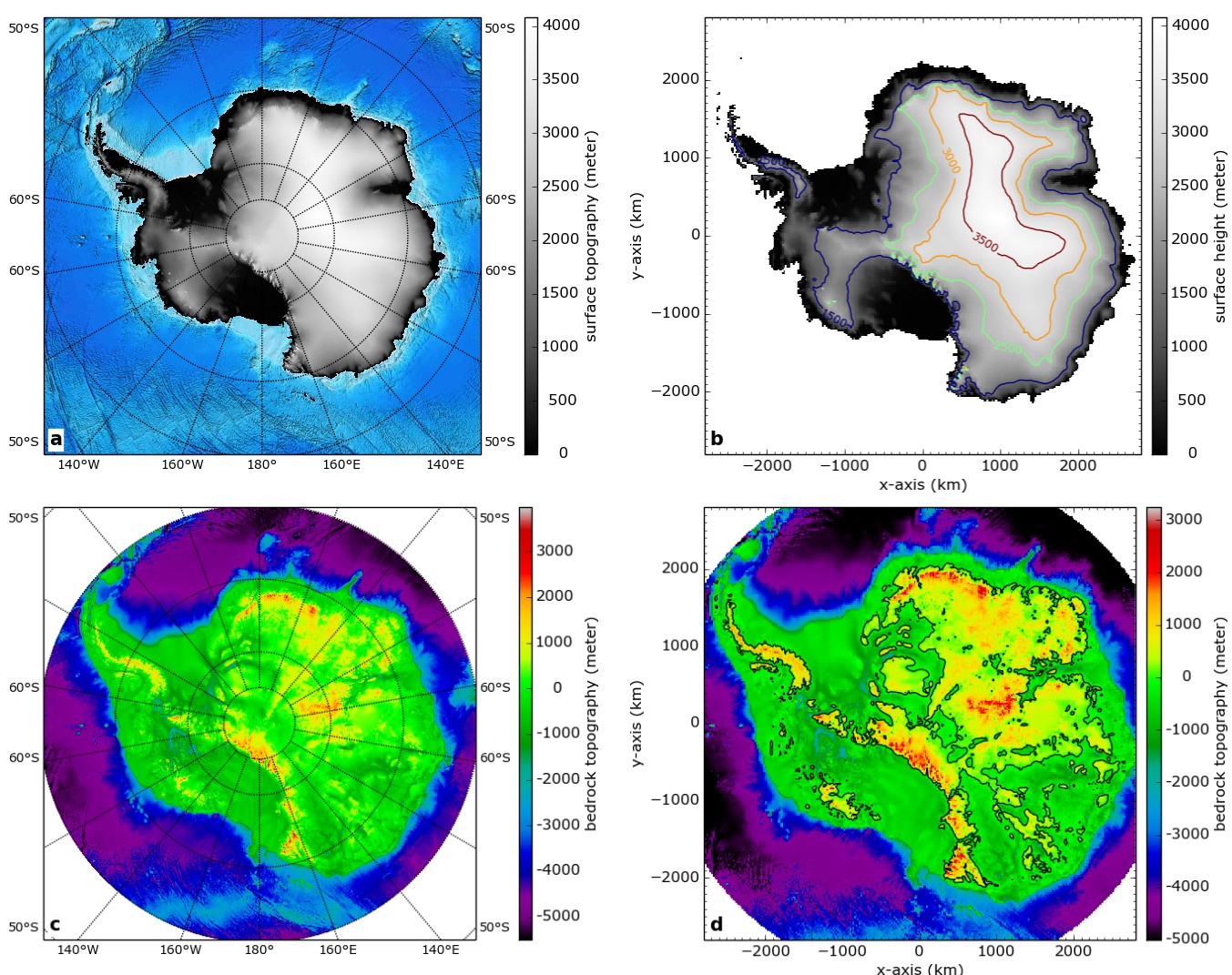

**Figure 9.** As Fig. 7 but for Antarctica, the Antarctic ISM grid resolution in **b** and **d** is 20x20 km.

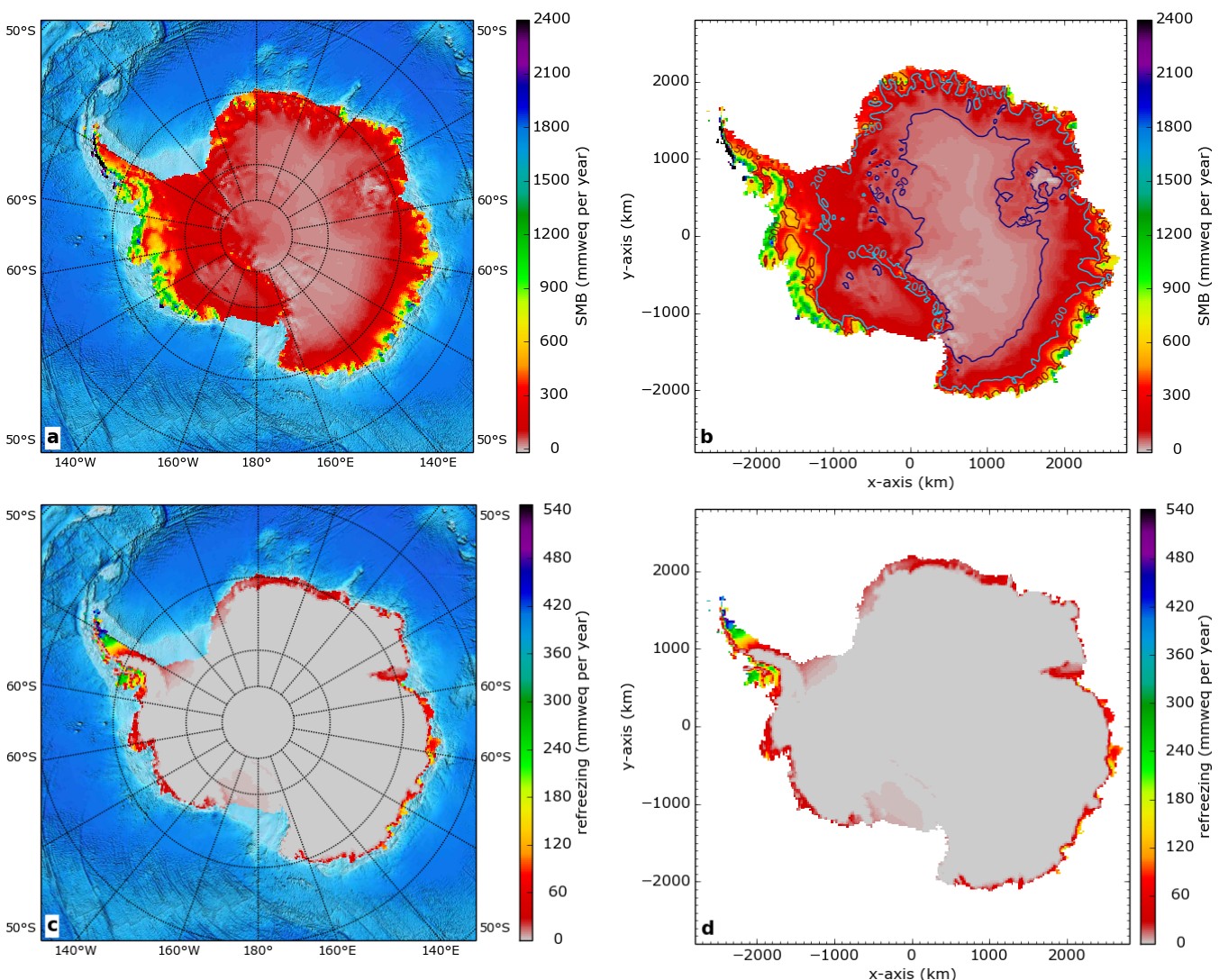

**Figure 10.** As Fig. 8 but for Antarctica, the Antarctic RACMO2.3 data set is a time average over the 1979–2014 period and the Antarctic RACMO2.3 grid resolution in **a** and **c** is about 27 km and the Antarctic ISM grid resolution in **b** and **d** is 20x20 km.

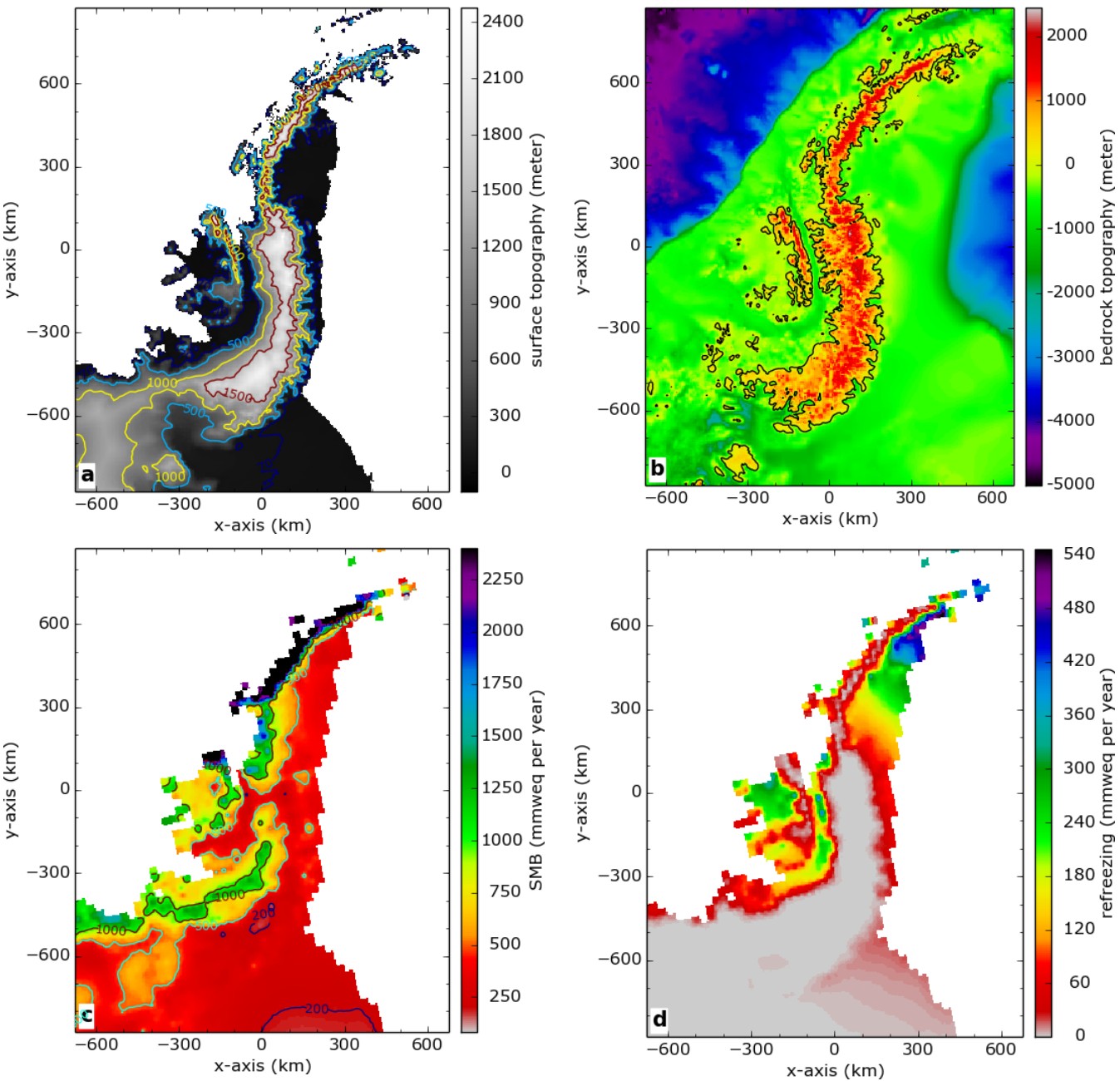

**Figure 11.** The Bedmap2 surface topography (Fig. 9a) and the bedrock topography (Fig. 9c) have been mapped on a local 5x5 km ISM grid for the Antarctic Peninsula with an optimal centred oblique projection without using a mask (see **a** and **b**). In **a** points at sea level have been plotted white in order to visualize the coastline contours. The 1979–2014 time-averaged RACMO2.3 SMB (Fig. 10a) and the refreezing (Fig. 10c) have been masked mapped on the same local 5x5 km ISM grid with the same optimal centred projection (see **c** and **d**).

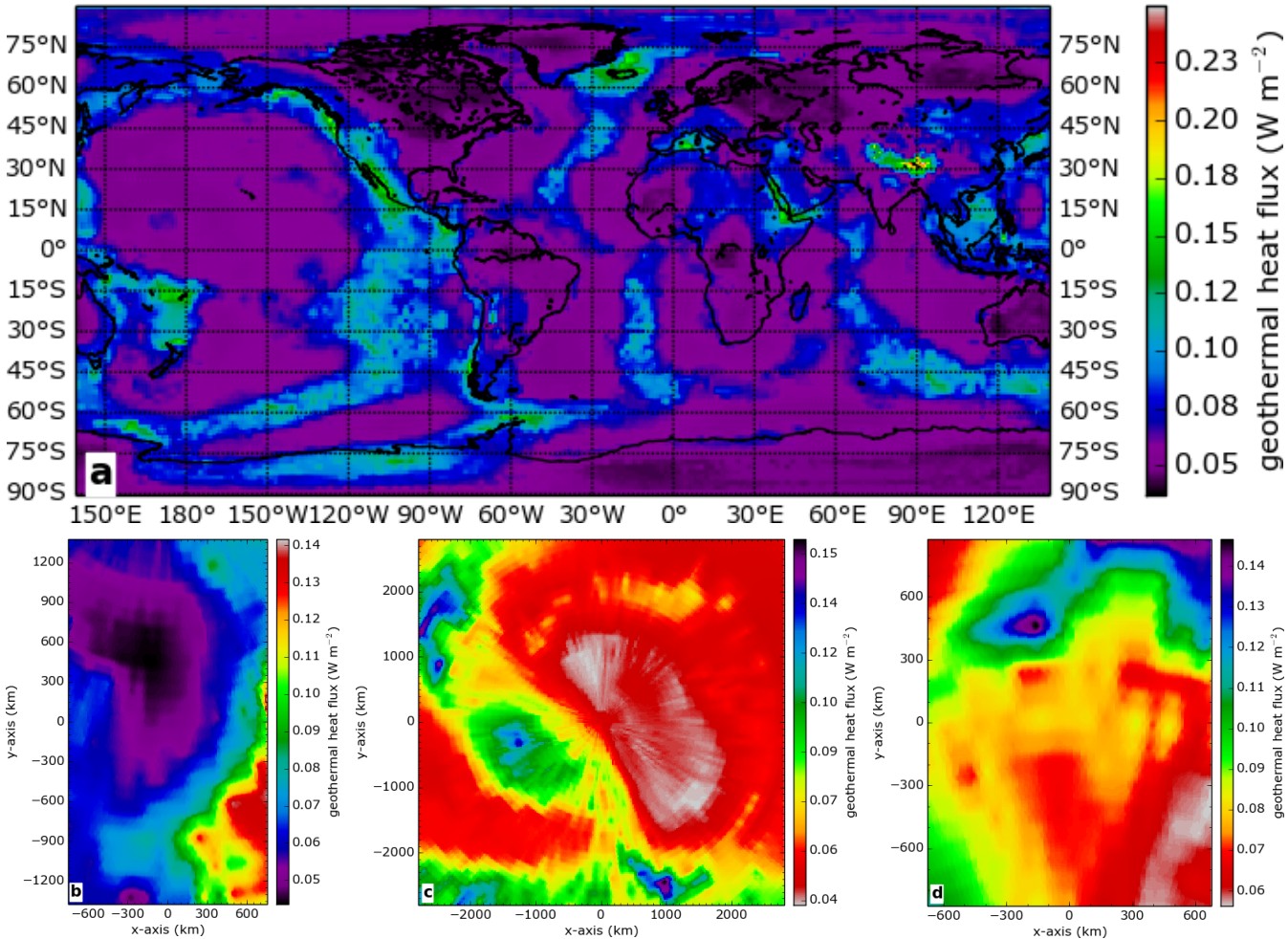

**Figure 12.** Subfigure **a** shows the geothermal heat flux (Shapiro and Ritzwoller, 2004) on a global regular $1° \times 1°$ longitude - latitude grid, the continental contours are plotted black. This geothermal heat flux has been mapped with optimal centred projections on a 5x5 km ISM grid for Greenland (**b**), a 20x20 km ISM grid for Antarctica (**c**) and a 5x5 km ISM grid for the Antarctic Peninsula (**d**).