# Peer review of "Fast GCM - ice sheet model coupling software OBLIMAP 2.0, including on-line embeddable mapping routines"

_Geoscientific Model Development, 2016_

## Referee Comment (RC1) · Anonymous Referee #1 · 29 Jun 2016

Reerink and van de Wal describe an updated version of their OBLIMAP mapping tool. This tool is designed for mapping fields between coarse-resolution GCM grids and fine-resolution ice sheet grids. The authors described OBLIMAP v. 1 in a 2010 GMD paper; this new paper describes updates made in v. 2 of this tool. Major new features include (1) a "fast scan" method that significantly improves the performance of the initial mapping step; (2) handling masked fields in the mapping; and (3) the capability to embed this tool within a GCM.

Based on citations of the original paper, it appears that v. 1 of this tool has proven to be quite useful to their research group. Indeed, it does sound like a useful tool, and it seems that v. 2 makes the tool even more convenient and generally useful, such

as through robust handling of masked fields. I read this paper with interest because I am currently struggling with issues surrounding conservative online regridding of fields passed between a GCM and ice sheet model. However, in the end I was not convinced that OBLIMAP v. 2 has sufficient novelty or improvements relative to alternatives to warrant its publication in GMD.

It is hard for me to tell how much this is a real issue vs. simply a presentation issue. In particular, the authors do not spend much time comparing this tool to other alternatives, so it was hard for me to tell what practical improvements it provides over alternatives. They briefly compare it with the OASIS coupler (p. 17, L. 7-14), although not in enough detail to convince me of its advantages. For example, what are the practical advantages of the additional projection step in OBLIMAP (p. 17, L. 8-9)? I would also like to see comparisons with other tools that serve a similar purpose, such as ESMF's regridding tools (https://www.earthsystemcog.org/projects/regridweightgen/, https://www.earthsystemcog.org/projects/esmf/regridding_esmf_6_3_0rp1), and tools that have been written on top of those in NCL and python.

The fast scan method was the most intellectually interesting component of this paper. I am not familiar enough with mapping tools and techniques to know if this is a novel contribution to the field, and/or if it might be generally applicable to other mapping tools. If it is, then this aspect could make the paper worth publishing. However, I was left with a feeling of uncertainty about the conditions under which the fast scan was guaranteed to give identical results to the full scan, the conditions under which it would "likely" give identical results, and the conditions under which it is not expected to give identical results. I got the impression that the authors themselves are not confident of the situations under which it is guaranteed to work correctly, based on vague statements like, "Even if the additional dynamic block size method is omitted, the \*majority\* of the fast scan mappings yield identical results with the full scan method. However, including it \*appears to be very effective\* in obtaining identical results for the exceptional cases." (p. 7, L 19-21; emphasis mine). If this is published, then I would like to see these conditions made more clear. For example, is it guaranteed to work for regionally-refined grids? And is the tool able to detect when the fast scan will or won't work, or is the burden on the user to determine this? Also, in order for this technique to be more re-producible, I would like to see more precise descriptions in place of some of the vague statements such as, "an ample rounded b is taken" (p. 7, L. 1-2).

I was especially interested in the new capabilities of this tool for online coupling be-tween a GCM and an ice sheet model. However, again, I was not convinced of the suitability of OBLIMAP for this purpose. As a GCM developer, I would need to be convinced that OBLIMAP provides enough benefits over the GCM's existing coupling technologies (e.g., Valcke et al. 2012) that it's worth using this additional package for regridding to/from the ice sheet grid. The authors did not convince me that this is the case. Again, it's hard to tell how much of this is merely a presentation issue: I would reconsider this paper for publication if the authors provided more detailed discussions of OBLIMAP's advantages. However, it does not appear that OBLIMAP provides the capability to handle either of the two major challenges that we are currently struggling with in ice sheet coupling: (1) regridding from coarse to high resolution grids in a con-servative but relatively smooth fashion, and (2) handling multiple elevation classes in the GCM – i.e., including vertical as well as horizontal remapping. Regarding (1), the authors state that the scheme is "close to conservative" (p. 2, L. 22), but do not quantify this. For coupled model simulations, a fully conservative mapping scheme is needed.

In terms of the actual mapping algorithms implemented by OBLIMAP, my sense is that "quadrant" interpolation is similar to bilinear interpolation, and "radius" interpolation is a blend between bilinear and area-conservative interpolation. However, it was hard for me to tell for sure. The authors should provide an overview of these two interpolation methods, and a brief comparison with other commonly-used interpolation methods. It was unclear to me why OBLIMAP uses these methods.

Additional specific comments follow:

1. The Introduction gets into too many details on projections, etc. Instead, I would want the Introduction to provide a high-level introduction to the problem, including a wider survey of other tools that serve a similar purpose. There could then be a separate section describing the details and limitations of OBLIMAP v. 1, which would include some of these details on projections.

2. p 9 L 1-5: It would be helpful if the API of OBLIMAP were laid out more explicitly here – what does the code interface look like in detail? This isn't absolutely necessary, but would help GCM developers evaluate whether they want to include OBLIMAP in their system.

3. It would also be helpful if an example were given of how to configure OBLIMAP for offline regridding (e.g., an example configuration file and/or command-line usage). Again, this isn't necessary, but would help people decide if this is something they want to use.

4. Section 4.2 and associated figures could be trimmed substantially. It seems that, relative to OBLIMAP v. 1, the main addition here is the masking, which could be illustrated with a single figure

5. Section 4.2 should be expanded to add metrics on area-integrated sums, to help judge conservation. My sense is that these schemes are fundamentally non-conservative, which would be a problem for incorporating them in a GCM.

6. p. 15 L 21-22: "The weaker second condition...": I don't understand this; the authors should reword to make this more clear.

7. There are numerous typos and grammatical errors. Just a few examples are:

a. p 1 L 7: "frequent" should be "frequency"

b. p 2 L 5: "constrains" should be "constraints"

c. There should not be an apostrophe in GCMs or ISMs

[Figure]

**References**

Valcke, S., Balaji, V., Craig, A., DeLuca, C., Dunlap, R., Ford, R. W., Jacob, R., Larson, J., O'Kuinghttons, R., Riley, G. D., and Vertenstein, M.: Coupling technologies for Earth System Modelling, Geosci. Model Dev., 5, 1589-1596, doi:10.5194/gmd-5-1589-2012, 2012.

---

## Referee Comment (RC2) · F. COLLEONI (Referee) · 29 Jun 2016

In this manuscript, Reerink et al. present an updated improved version of the interpolation tool OBLIMAP whose aim is to interpolate the GCM fields onto ice sheet models grid and vice-versa. I must say that I am myself a regular user of the first version of OBLIMAP since many years now. However, many of the ice sheets models that I use have grids projected on the ellipsoid and not on the sphere. Therefore at the time of OBLIMAP 1.0, I had to implement by myself the missing projection routines on ellipsoid grids. In addition, because I also needed to interpolate ocean vertical data, or create my own ISM regional grid, I also had to implement the additional level dimension in the netcdf routine and produce a slightly modified version of the first version of OBLIMAP to

create new ISM grids for boundary conditions. Recently, I needed to use the BEDMAP Antarctic topography, which comes in cartesian format already projected and the problem of re-interpolating it on a different grids surged. In this new version of OBLIMAP, all the points that I mentioned above have been fully improved and implemented, which demonstrate that those points were really the main weaknesses of the first version.

Several improvements that have been made in this version: it provides the geographical projections on ellipsoid and not only on a sphere as in the previous version. In fact, many ice sheet models have their cartesian grid projected on the ellipsoid and not on the sphere. This was a limiting aspect of the first version since one had to implement the projection on the ellipsoid by himself. the computation time performance of the scan method to interpolate from grid with numerous points has improved substantially. This was effectively a limitation. A suggestion could also be to provide a parallel version of this code for this specific loop. the re-interpolation of cartesian projected topographic or climate datasets on a defined ISM cartesian grid. Which is to me, one of the greatest improvements. it can maps 4D fields, such as ocean or winds, which is highly appreciable improvements.

OBLIMAP is a really useful tool for those who carry out ice sheet simulations and the difficulty of interpolating on cartesian grids can be understood only when facing this problem. On the coupling process, I would definitely say that OBLIMAP should be embedded in ice sheet models rather than in the coupler of coupled climate models. Given the variety of existing climate couplers, and sometimes their complexity (e.g. the NCAR model), it is much easier to host OBLIMAP within the ice sheet models. As I mentioned again in the Major comments below, the GLIMMER ice sheet model and I think also the CISM model (derived from GLIMMER), the TARAH ice sheet model (Pollar ad Deconto) already embed the projection of the climate fields onto the ice sheet grid within the ice sheet code itself, and by experience with both ice sheet and climate models, I would also recommend to put OBLIMAP in the ISM code.

Based my comments above, except minor typos or reformulations, I recommend the

publication of the manuscript in its current shape.

Florence Colleoni

Major comments

Maybe a stand-alone version, as for the first version could also be useful, not everybody uses coupled system. Or maybe this version is stand-alone but this is not clear and should be clarified with a sentence or so in manuscript. Line 50-55: after using OBLIMAP very often, I would say that perhaps hosting OBLIMAP in the ice sheet model is the most easiest way to deal with it. OBLIMAP is a small coupler model and very simple in its use, therefore, it is easily implemented in an ISM, as for example GLIMMER does, rather than embedded in a GCM. But this is only my opinion of user. In addition the climate coupler are built to interpolate on lon-lat grids most of the time.

Minor comments

line 45: substitute "albedo changes" by "ice sheet distribution" because an ISM does not provide albedo changes, it only provides ice distribution which affect albedo within the atmosphere model. A regional atmospheric model as RACMO on the contrary provides albedo changes, but this is not an ISM.

line 48: "surface mass balance" put a space between "mass" and "balance"

line 52: substitute "ocean surface temperatures" by "ocean temperatures.". Most of the basal melting methods uses vertical ocean temperature and salinity distribution (Holland and Jenkins 1999, Pollard and Deconto 2012, Martin et al., 2011).

line 63: substitute by "any regional energy balance model (e.g. RACMO or MAR)"

Figure 7: "surface mass balance" put a space between "mass" and "balance"

---

## Author Comment (AC1) · 23 Aug 2016

**Answer to Referee 1**

First we would like to thank the referee for his/her comments on our work which improved our manuscript. We carefully considered your comments. You will find a detailed answer to each of your comments below. We hope that the revisions improve the quality of the paper and meet your expectations.

Reerink and van de Wal describe an updated version of their OBLIMAP mapping tool. This tool is designed for mapping fields between coarse-resolution GCM grids and

fine-resolution ice sheet grids. The authors described OBLIMAP v. 1 in a 2010 GMD paper; this new paper describes updates made in v. 2 of this tool. Major new features include (1) a "fast scan" method that significantly improves the performance of the initial mapping step; (2) handling masked fields in the mapping; and (3) the capability to embed this tool within a GCM.

Based on citations of the original paper, it appears that v. 1 of this tool has proven to be quite useful to their research group. Indeed, it does sound like a useful tool, and it seems that v. 2 makes the tool even more convenient and generally useful, such as through robust handling of masked fields. I read this paper with interest because I am currently struggling with issues surrounding conservative online regridding of fields passed between a GCM and ice sheet model. However, in the end I was not convinced that OBLIMAP v. 2 has sufficient novelty or improvements relative to alternatives to warrant its publication in GMD.

It is hard for me to tell how much this is a real issue vs. simply a presentation issue. In particular, the authors do not spend much time comparing this tool to other alternatives, so it was hard for me to tell what practical improvements it provides over alternatives. They briefly compare it with the OASIS coupler (p. 17, L. 7-14), although not in enough detail to convince me of its advantages. For example, what are the practical advantages of the additional projection step in OBLIMAP (p. 17, L. 8-9)?

We first have to point to the fact that the projection step is an essential obligatory step in case two models run on differently curved surfaces. So the discussion is not about *"what the practical advantages are of the additional projection step"*, because one has to (inverse) project the surface and thus the coordinates of the grid nodes if the surfaces differ in curvature, which is the case when a GCM which runs on the Earth Sphere surface is coupled with an ISM which runs on a flat surface. This is different when regridding two ESM components which both run on the same Earth Sphere surface. In that case a projection is not needed as one stays on the same curved surface and only the interpolation step is required to regrid.

This additional (inverse) projection step in GCM – ISM coupling has a few important consequences for the cross grid search (the method to select at a given destination grid point the contributing destination points which are used for the interpolation). Due to the projection, it is in general a priori unknown how the grid nodes of the two grids are related to each other, the projected nodes can end up anywhere depending on the projection. The search method or as we call it the 'scan method' has to robustly cope with that. Other typical different requirements in GCM – ISM coupling compared to ESM component coupling are: (1) The ISM grid concerns a local part of the GCM which requires a neat treatment of this mapped ISM domain border. (2) If the local mapped ISM field is mapped on the GCM this result has usually to be merged into the existing GCM field. (3) The range of resolution ratios is much larger, i.e. often the ISM grid resolution is much finer than that of the GCM. These rather specific requirements are the cause that GCM – ISM coupling is not standard included in the existing ESM component couplers.

OBLIMAP adresses these specific GCM – ISM coupling issues. Whereas the other mentioned couplers are hubs from which several ESM components are coupled and which include the sphere to sphere regridding. So it is not that we can discuss *"its advantages"* over couplers like OASIS, the ESMF coupler, the CPL6 coupler, the CPL7 coupler, the MCT-based couplers or the C-Coupler, as they serve different targets. Nevertheless, there are many functional similarities between these couplers and OBLIMAP. Moreover, beside the option to embed the OBLIMAP routines in the GCM or ISM, in fact another option is to call an ISM with OBLIMAP embedded from one of the ESM component couplers. OBLIMAP itself is not that much a hub from which more than two ESM components are coupled. However for the two components, the GCM and the ISM, we expect that the design compares/integrates well with these ESM component couplers.

Having said that the projection step is an inevitable step for GCM – ISM coupling and that OBLIMAP is a specific GCM – ISM coupler, we checked our introduction and the

discussion (p. 17, L. 7-14) whether this is clear. We think that the first issue is clearly addressed at (p. 1, L. 16-18) and (p 2. L. 1-3), but have more emphasized this point in the abstract and the start of the discussion. However, we think that we can be more specific about the different purposes which are served by OBLIMAP and the other couplers like OASIS. Also more couplers can be mentioned and compared. This will be improved by a rewrite/addition of the first paragraph of the introduction and of the discussion (p. 17, L. 7-14).

I would also like to see comparisons with other tools that serve a similar purpose, such as ESMF's regridding tools (https://www.earthsystemcog.org/ projects/ regridweight-gen/, https://www.earthsystemcog.org/ projects/ esmf/ regridding _esmf_6_3_0rp1), and tools that have been written on top of those in NCL and python.

We are only able to compare the ESMF_RegridWeightGen routine based on the specification provided on their website, we found no other publication than Valcke (2012) and Liu et al. (2014) which shortly summarize several couplers. We studied chapter 12 and 23.2 of the ESMF reference documentation (Version 6.3.0rp1), but both sections seem to be not always mutual consistent. For example Sect. 12.1 mentions that the source and destination grid must be on the sphere, but Sect. 23.2 seems to suggest that the routine also works in case both grids are on the flat plane. In either case this is different from OBLIMAP, in OBLIMAP one of the grids coincides with the surface of the sphere or the ellipsoid while the other one represents the flat surface (or both represent the flat surface). Secondly, as far we can judge from this description OBLIMAP has a much more flexible and extended masking facility, which is entirely independent of the scan phase (the preceding offline weight factor generation). Thirdly, OBLIMAP actually does not store the weight factors in the SID file like SCRIP does, but the indices of the contributions and their distances which offers the flexibility in a post scan phase to change the mask and the distance weighting exponent. Note that zero entries, which can be a large amount depending on the participation mask, are not stored in the SID file. The SID file and DDO have been designed such that the required processor memory is

minimized. Based on this design a well scalable parallel bitwise-identical implementation is expected to allow further future performance improvements. OBLIMAP does not use the matrix multiplication. Due to a possible large resolution ratio the sparse matrix could have a rather large amount of non-zero diagonals. Instead OBLIMAP uses the direct access via the DDO (which has to be loaded only at initialization) to the indices and distance of the contributions, which allows a very fast evaluation.

The ESMF_RegridWeightGen uses the SCRIP (Jones 1999) grid and weight file formats, it seems very likely that the package is largely based on SCRIP probably with some modifications (and extensions for flat surfaces) but this is not mentioned. Because the OASIS, CPL6 and CPL7 couplers also use SCRIP for generating the interpolation weights for the regridding, the comparison of OBLIMAP with one of them applies to all.

Parts of this discussion are added to the discussion section. And the difference between SCRIP's regridding for spherical coordinates and OBLIMAP (dealing with different curved surfaces) is now addressed at the begin of the introduction.

The fast scan method was the most intellectually interesting component of this paper. I am not familiar enough with mapping tools and techniques to know if this is a novel contribution to the field, and/or if it might be generally applicable to other mapping tools. If it is, then this aspect could make the paper worth publishing. However, I was left with a feeling of uncertainty about the conditions under which the fast scan was guaranteed to give identical results to the full scan, the conditions under which it would "likely" give identical results, and the conditions under which it is not expected to give identical results. I got the impression that the authors themselves are not confident of the situations under which it is guaranteed to work correctly, based on vague statements like, "Even if the additional dynamic block size method is omitted, the \*majority\* of the fast scan mappings yield identical results with the full scan method. However, including it \*appears to be very effective\* in obtaining identical results for the exceptional cases." (p. 7, L 19-21; emphasis mine). If this is published, then I would like to see these conditions made more clear. For example, is it guaranteed to work for regionally-refined grids? And is the tool able to detect when the fast scan will or won't work, or is the burden on the user to determine this? Also, in order for this technique to be more reproducible, I would like to see more precise descriptions in place of some of the vague statements such as, "an ample rounded b is taken" (p. 7, L. 1-2).

The referee asks to clarify the conditions under which the fast scan is guaranteed/applicable. We have rewritten the second part of the fast scan section (p. 6, L. 25 upto p. 7, L. 32) to make this conditions easier to understand and improved the description of the method. We replaced our former general restriction by stating that the fast scan method is applicable to structured grids (synonyms for structured grid are curvilinear grid or logically rectangular grid). A short subsection is inserted in Sect. 3.1 to define the terms structured and unstructured grids. Other parts in the text are adjusted as well to this terminology.

Regionally-refined grids do not satisfy the structured grid conditions, this is addressed in the revised text. The default fast scan method is robust for all structured grids. The OBLIMAP-package includes a verification script to verify situations in which a less robust but faster scan method is tested or if the method is tested for certain unstructured grids. The supplement (the OBLIMAP User Guide) describes this verification. The discussion shortly addresses a possible solution to make the fast mapping applicable for unstructured grids.

I was especially interested in the new capabilities of this tool for online coupling between a GCM and an ice sheet model. However, again, I was not convinced of the suitability of OBLIMAP for this purpose. As a GCM developer, I would need to be convinced that OBLIMAP provides enough benefits over the GCM's existing coupling technologies (e.g., Valcke et al. 2012) that it's worth using this additional package for regridding to/from the ice sheet grid. The authors did not convince me that this is the case. Again, it's hard to tell how much of this is merely a presentation issue: I would reconsider this paper for publication if the authors provided more detailed discussions

of OBLIMAP's advantages.

*As pointed out in our first comment the discussion is not about OBLIMAP's benefits or advantages over the GCM's existing coupling technologies*, because they serve different targets. Because OBLIMAP is a specific GCM – ISM coupler with a sound performance (Reerink et al. 2010) and because its current release includes many extended user options (see the OBLIMAP User Guide for the full list), has an efficient design which enables embedded use, installs easy and is easy to configure, we think the current release is convincingly beneficial for users. We revised the manuscript to emphasis its specific GCM – ISM coupling tasks and widened the comparison with the ESM component couplers.

However, it does not appear that OBLIMAP provides the capability to handle either of the two major challenges that we are currently struggling with in ice sheet coupling: (1) regridding from coarse to high resolution grids in a conservative but relatively smooth fashion,

At several points in this review the issue of conservative mapping is addressed. We agree that conservation is a very relevant theme in mapping. However, to which extent a mapping has to be conserved could be subject of a discussion. We will discuss here several issues.

First we note that the ESM component couplers OASIS, CPL6, CPL7 and ESMF rely on SCRIP (Jones 1999). SCRIP is able to remap from sphere to sphere, the typical task for ESM component couplers. Jones (1999) shows that the remapping is much less accurate (especially for fields with large gradients) for first order area conservative interpolation than for e.g. bilinear interpolation. Jones (1999) therefore presents a more accurate second order area conservative interpolation. However, none of the ESM component couplers use the second order variant, all of them use the first order area conservative remapping. We think the reason for that is that the second order variant needs the gradient of the field, which is problematic because this does not allow

prior offline generation of the interpolation weights, and would be field dependent. And, if we understand correctly, the areas are approximated by the line segments over which the line integrals are carried out for irregular overlapping grids, which makes it not exact area conservative.

The area distortion due to the projection is another complication if conservative interpolation in GCM – ISM coupling is considered. For a very local projection the LAEA projection could be used, but for larger scale ice caps it is important that (flow) directions are not affected by the projection in order to stay close to the physical representation of the models. This means that in general a SG projection deforms the area of each cell. The combination of a projection with an area conservative mapping leads to large errors. If e.g. the area of a cell shrinks by 1%, the value of that cell will increase by 1% to compensate due to the area conservation. However, the area mismatch is compensated after the reverse mapping, therefore the conservation of the GCM – ISM coupling should be judged by comparing the results after a to and fro mapping. This requires adequate tests, like those carried out by Reerink et al. (2010) which show results close to conservation. The quadrant and radius interpolation method which are based on the inverse squared distance weighting perform well with respect to conservation after to and fro mapping (Reerink et al. 2010). OBLIMAP uses the radius method to obtain a representative estimate for mapping from coarse to high resolution grids.

We think that exact area conserved mapping should be not too much at the expense of the accuracy of the mapping, at least not for the fields which are involved in GCM – ISM mapping. Hence a balance has to be found between optimal accuracy and a conserved behaviour in the coupling. As we have to deal with the projection in GCM – ISM coupling, our strategy has been to reduce the area distortion. This is achieved by using oblique projections and is optimized by taking an optimal standard parallel. For less local mappings we prevail accuracy in mapped directions in order to stay close to the ice flow physics, instead of exact area conservation. However, compared over the the to and fro mapping Reerink et al. (2010) show that OBLIMAP maps close to
conservative.

We added a paragraph to the discussion to discuss this.

and (2) handling multiple elevation classes in the GCM – i.e., including vertical as well as horizontal remapping.

OBLIMAP 2.0 includes the mapping of spatial 1D, 2D and 3D fields with or without a time dimension. Like the C-Coupler (Liu et al. 2014) it concerns a 2D + 1D mapping for 3D fields, in the sense that the horizontal mapping includes the 2D interpolation with the distance weighting (with the prior scan option). Each vertical layer (or elevation class level) is treated with the same 2D horizontal interpolation but is not interpolated in the vertical direction by OBLIMAP. Returning the vertical layers just as vertical records is a conscious choice, it keeps the most flexibility. For example it allows the vertical coordinate to change without affecting the mapping, i.e. avoiding a repeated scan phase. This is particular important regarding the vertical zeta coordinate in ISM models which usually not only changes in time but even changes per grid node in time (a rather weird phenomenon for non-ISM modellers). In this way the vertical grid is allowed to match with either a real or scaled coordinate and could differ per field, again without affecting the mapping. It allows direct downscaling if one wishes, which in that case saves one interpolation step. This is all possible without losing much on the performance, because the vertical interpolation is computational straightforward and at low cost. Furthermore, it is independent of a possible future parallel domain decomposition as noticed by Liu et al. (2014). An added paragraph to the discussion addresses this remarks, and small adjustments have been made to Sect. 3.6.

Regarding (1), the authors state that the scheme is "close to conservative" (p. 2, L. 22), but do not quantify this. For coupled model simulations, a fully conservative mapping scheme is needed.

This *close to conservative (p. 2, L. 22)* addresses the projection part, minimizing the distortions improves conservation. This is achieved by using an optimal aligned oblique

projection. An extensive quantification of the nearly conserved to and fro mapping is presented in Reerink et al. (2010). We changed the text at (p. 2, L. 27) to refer explicitly to the conservation tests in Reerink et al. (2010).

In terms of the actual mapping algorithms implemented by OBLIMAP, my sense is that "quadrant" interpolation is similar to bilinear interpolation, and "radius" interpolation is a blend between bilinear and area-conservative interpolation. However, it was hard for me to tell for sure. The authors should provide an overview of these two interpolation methods, and a brief comparison with other commonly-used interpolation methods. It was unclear to me why OBLIMAP uses these methods.

Although Sect. 2.3 in Reerink (2010) provides a straightforward and sound description of the quadrant and radius interpolation methods (to which is referenced (p. 2. L. 17) in the introduction), we agree with the referee that the motivation to choose these interpolation techniques could receive more attention. We therefore inserted the answer provided below as a section 3 of the updated supplement, and rewrote the paragraph (p.2, L. 15–19) in the introduction.

The quadrant and radius method are both inverse squared distance weighting interpolation methods based on Donald Shepard's famous paper 'A two-dimensional interpolation function for irregularly-spaced data' which he published in 1968. His introduction discusses the shortcomings of the bilinear (called double linear), bicubic and other interpolation methods if applied to irregularly-spaced data.

The inverse squared distance weighting function has a few very practical advantages when interpolating spatial data, it is suited to identically treat: (1) regular and irregular spaced grid nodes, (2) 1D, 2D and 3D spatial grids, (3) any curved destination surface, i.e. the surface of a sphere and ellipsoid or the flat surface, (4) any number of weighting contributions. The weighted average is based on weighting the inverse squared distances of all the selected contributions. Although the weighting factors are default taken equal to the inverse *squared* distance as recommended by Shepard, it is

possible (also in OBLIMAP) to replace this exponent = 2 which makes it *squared*, by each exponent equal or larger than 1 instead.

This pure inverse squared distance weighting function has to be combined with a limiting influence distance in order to select only nearby points, as described by Shepard. The main reason for this limitation is the computational performance, but it also avoids the need of suppressing a possibly biased average in case there is a relative large number of distant points involved.

As noted by Shepard there are many ways to limit the number of contributions. In OBLIMAP we implemented two methods to limit the influence of distant contributions, based on three typical situations encountered by mapping: (1) a coarse grid is mapped on a fine grid, (2) a grid is mapped on a grid with a similar resolution, (3) a fine grid is mapped on a coarse grid.

The first and second situation are addressed by OBLIMAP's default interpolation method, the quadrant interpolation method, which draws a cross through the considered destination point, and selects in each quadrant the nearest projected contribution. It is a relative arbitrary choice to divide the surrounding area in four segments, in fact it could be divided in any number of segments. The choice for four segments is slightly inspired by the bilinear interpolation which also uses four surrounding points. This selection method does effectively shadow other contributions in the same segment/direction in a simple way. Note that with an increasing number of segments the shadowing becomes more direction sensitive.

In the third situation, in which a fine grid is mapped on a coarse grid, OBLIMAP uses the radius interpolation method which selects those contributions which lay within a certain radius. A reasonable radius typically equals half the departure grid size resolution. The basic idea is that the coarse destination grid cell obtains a representable average value. Because the number of selected contributions increases approximately squared with an increasing selecting radius (given a constant node density), more distant points are

selected but they weight inversely squared. This squared and inverse squared effect compensates and makes that the radius method generates a representable average estimate.

We preferred selecting within a radius over selecting the $n$ nearest points which is another well known method, because the latter requires sorting which is notorious computational expensive for large $n$ and complicates the interpretation of the results in situations with masks and data gaps, and also does not directly match the area size of the destination grid cell.

As the weighting function itself just weights over the number of detected contributions, segments are allowed to stay empty. Therefore the method is robust for destination grid domain edges, mapped departure grid domain edges, data gaps, and masked points where the mask is also allowed to differ per field and per vertical layer, and all these different masks are even allowed to change in time.

In both interpolation methods the distances between the considered destination point and a projected departure point are calculated over the destination surface along the great circle. Available surface curvatures in OBLIMAP are the surface of a sphere, an ellipsoid and of a flat plane.

Default OBLIMAP uses the quadrant interpolation method, but the radius interpolation method is automatically selected if the resolution of the destination grid is four times coarser than the resolution of the departure grid. The interpolation can be configured manually as well, for that and for all defaults see the OBLIMAP User Guide.

Independent of the interpolation method which has been used in the scan phase, the nearest point assignment can be used in the post scan phase and will match with masks which change in time and differ per field and layer.

Bilinear interpolation requires exception rules if situations are encountered with less than four contributions, and becomes a less well defined method for irregularly spaced

contribution points. Bilinear interpolation is not the same as the quadrant interpolation. In case a regular Cartesian grid is remapped, the bilinear interpolation weights a contribution based on the surface of the opposite corner (see e.g. the third figure at https://en.wikipedia.org/ wiki/ Bilinear _interpolation), while the quadrant interpolation weights this contribution based on the inverse squared distance. For a 1 by 1 unit grid with a certain contribution at distance $(\Delta x, \Delta y)$ from destination point P, the weighting factor of this contribution to P is $\Delta x^2 + \Delta y^2$ for the quadrant interpolation instead of $(1 - \Delta x)(1 - \Delta y)$ for the bilinear interpolation. The normalizing factors, which also differ per method because they are a summation over the weights of all participating contributions, are omitted here.

The radius interpolation has no real similarities with the bilinear interpolation. As explained above the squared contribution covering area and the inverse squared distance weighting of each contribution compensate. This generates an estimate which represents an area average close to conservative as all local parts of the cell are in fact treated equally with respect to area weighting.

**Additional specific comments follow:**

1. The Introduction gets into too many details on projections, etc. Instead, I would want the Introduction to provide a high-level introduction to the problem, including a wider survey of other tools that serve a similar purpose. There could then be a separate section describing the details and limitations of OBLIMAP v. 1, which would include some of these details on projections.

As pointed out earlier, the projection is an essential and distinguishing part of the GCM – ISM coupling. We think that the relevant part of OBLIMAP's first release should be summarized in the introduction. The projection description is an important part of that. We agree to widen the comparison with more couplers and to emphasis the difference between a GCM – ISM coupler like OBLIMAP and the other ESM component couplers. This will be addressed at the begin of the introduction and in the discussion.

2. p 9 L 1-5: It would be helpful if the API of OBLIMAP were laid out more explicitly here – what does the code interface look like in detail? This isn't absolutely necessary, but would help GCM developers evaluate whether they want to include OBLIMAP in their system.

We agree that the reader is served by an more explicit description of the API, we will add a schematic code figure containing the OBLIMAP API indicating the location in the host model code. Some additional sentences discussing and referring to the figure will be added to this section.

3. It would also be helpful if an example were given of how to configure OBLIMAP for offline regridding (e.g., an example configuration file and/or command-line usage). Again, this isn't necessary, but would help people decide if this is something they want to use.

We regret that the link to the supplementary information (the OBLIMAP User Guide) was not properly inserted in the discussion paper. These examples, including an example of a config file, are present in the OBLIMAP User Guide. We will suggest that this OBLIMAP User Guide pdf will be directly linked, like a usual supplementary pdf. This makes the supplement easier accessible, and avoids that a reader first has to download the OBLIMAP-package and search therein.

4. Section 4.2 and associated figures could be trimmed substantially. It seems that, relative to OBLIMAP v. 1, the main addition here is the masking, which could be illustrated with a single figure

Section 4.2 demonstrates that OBLIMAP is a powerful tool which is able to map diverse kinds of topographic and forcing data sets onto any ISM grid configuration with an optimal oblique projection. The public available high resolution topographic data sets are remapped (reprojected from a polar to an optimal oblique aligned projection for the ellipsoid) for each area on a certain ISM grid of preference, i.e. with the desired grid extensions and grid resolution. The atmospheric forcing data sets which are defined

on a reduced gaussian grid of the regional RACMO2 model, are mapped from the sphere to the same ISM grids. The geothermal heat flux field which is defined on a global regular longitude-latitude grid, is also mapped from the sphere to these same ISM grids. Besides, these different data sets cover a wide resolution range and map the two major ice sheets, in addition the Antarctic Peninsula example shows how a subregion is mapped with its own optimal oblique projection. Furthermore, masked mapping is shown for a relevant set of applications and different masking issues are discussed. We think that Section 4.2 shows the capabilities which OBLIMAP offers ice modellers, which we think is underlined by the comments made by the second referee. At the start of Sect. 4.2 the different purposes of this section are know summarized.

We realize, however, that these RACMO2 fields are not public available yet and the two data sets will therefore accompany the OBLIMAP 2.0 package, which will be a supplement on the GMD site.

5. Section 4.2 should be expanded to add metrics on area-integrated sums, to help judge conservation. My sense is that these schemes are fundamentally non- conservative, which would be a problem for incorporating them in a GCM.

We refer to the area conservative discussion above, and point to the fact that the mapping quality of OBLIMAP has been the topic of Reerink et al (2010). The metrics on area-integrated sums are addressed at the end of Sect. 4 and in Table 4 in Reerink et al (2010).

6. p. 15 L 21-22: "The weaker second condition. . .": I don't understand this; the authors should reword to make this more clear.

After the rewrite of the second part of the section which describes the fast scan method, we now distinguish between structured grids and unstructured grids. Therefore this part of the discussion is rewritten as well.

7. There are numerous typos and grammatical errors. Just a few examples are:

a. p 1 L 7: "frequent" should be "frequency"
b. p 2 L 5: "constrains" should be "constraints"
c. There should not be an apostrophe in GCMs or ISMs

We apologize for the typos and errors, we have corrected them.

References
Valcke, S., Balaji, V., Craig, A., DeLuca, C., Dunlap, R., Ford, R. W., Jacob, R., Larson, J., O'Kuinghttons, R., Riley, G. D., and Vertenstein, M.: Coupling technologies for Earth System Modelling, Geosci. Model Dev., 5, 1589-1596, doi:10.5194/gmd-5-1589-2012, 2012.

Jones, P.: First- and second-order conservative remapping schemes for grids in spherical coordinates, Monthly Weather Review, 127, 2204-2210, 1999.

Liu, L., Yang, G., Wang, B., Zhang, C., Li, R., Zhang, Z., Ji, Y., and Wang, L.: C-Coupler1: a Chinese community coupler for Earth system modeling, Geosci. Model Dev., 7, 2281-2302, doi:10.5194/gmd-7-2281-2014, 2014.

Reerink, T. J., Kliphuis, M. A., and van de Wal, R. S. W.: Mapping technique of climate fields between GCMs and ice models, Geoscientific Model Development, 3, 13-41, doi:10.5194/gmd-3-13-2010, http://www.geosci-model-dev.net/3/13/2010/, 2010.

Shepard, D.: A two-dimensional interpolation function for irregularly-spaced data, Proceedings-1968 ACM National Conference, pp. 517-524, 1968.

---

## Author Comment (AC2) · 23 Aug 2016

First we would like to thank the referee for her compliments on the improvements of the OBLIMAP package, and for her comments on our work which improved our manuscript. You will find detailed comments and answers below. We hope that the revisions improve the quality of the paper and meet your expectations.

In this manuscript, Reerink et al. present an updated improved version of the interpolation tool OBLIMAP whose aim is to interpolate the GCM fields onto ice sheet models

grid and vice-versa. I must say that I am myself a regular user of the first version of OBLIMAP since many years now. However, many of the ice sheets models that I use have grids projected on the ellipsoid and not on the sphere. Therefore at the time of OBLIMAP 1.0, I had to implement by myself the missing projection routines on ellipsoid grids.

We are pleased to notice that OBLIMAP has been regularly used by others as well. The final published release of OBLIMAP 1.0 actually does include the option to map on the ellipsoid. However, in OBLIMAP 2.0 all polar aspect forms for the ellipsoid projections are now included, and any ellipsoid shape can be configured from the config file now (the WGS84 is still the default).

In addition, because I also needed to interpolate ocean vertical data, or create my own ISM regional grid, I also had to implement the additional level dimension in the netcdf routine and produce a slightly modified version of the first version of OBLIMAP to create new ISM grids for boundary conditions. Recently, I needed to use the BEDMAP Antarctic topography, which comes in cartesian format already projected and the problem of re-interpolating it on a different grids surged. In this new version of OBLIMAP, all the points that I mentioned above have been fully improved and implemented, which demonstrate that those points were really the main weaknesses of the first version.

Indeed OBLIMAP 2.0 enables mapping of spatial 1D, 2D and 3D fields, all with or without a time dimension. As setting the dimensional properties got laborious for the user, we enabled an option to automatically detect the dimensional shape of the input fields in any situation in the new release.

The expressed appreciation of the remapping feature of ISM data did us realize that running this simple two step procedure should be described in the OBLIMAP User Guide. Therefore we added this as a subsection (so it is easy to find from the index) of the 'Running OBLIMAP' section in the OBLIMAP User Guide.

Several improvements that have been made in this version: it provides the geographical
projections on ellipsoid and not only on a sphere as in the previous version. In fact, many ice sheet models have their cartesian grid projected on the ellipsoid and not on the sphere. This was a limiting aspect of the first version since one had to implement the projection on the ellipsoid by himself. the computation time performance of the scan method to interpolate from grid with numerous points has improved substantially. This was effectively a limitation. A suggestion could also be to provide a parallel version of this code for this specific loop. the re-interpolation of cartesian projected topographic or climate data sets on a defined ISM cartesian grid. Which is to me, one of the greatest improvements. it can maps 4D fields, such as ocean or winds, which is highly appreciable improvements.

We agree that with the fast scan method barriers are razed, e.g. the (re)mapping of high resolution data, like the currently available 1x1 km topographic fields for Greenland and Antarctica, was simply impossible before. Other more 'common sized' mappings, which typically took a few minutes with OBLIMAP 1.0, take less than a second now, which is also convenient. However, a parallel implementation of the scan phase would certainly be beneficial. A start has been made: A proposal to work on a parallel scan phase of OBLIMAP for the Polar Science HPC Hackathon 2016 was accepted, so I went in July 2016 to XSEDE's conference in Miami and worked with HPC experts on that. A well scalable parallel domain decomposition could be implemented for the scan phase, the results remain bitwise identical for a changing number of processors. This is work in progress. Advancing insights due to this recent work are reflected in a rewrite of the discussed proposed parallel implementation at the end of the discussion.

OBLIMAP is a really useful tool for those who carry out ice sheet simulations and the difficulty of interpolating on cartesian grids can be understood only when facing this problem. On the coupling process, I would definitely say that OBLIMAP should be embedded in ice sheet models rather than in the coupler of coupled climate models. Given the variety of existing climate couplers, and sometimes their complexity (e.g. the NCAR model), it is much easier to host OBLIMAP within the ice sheet models. As I

mentioned again in the Major comments below, the GLIMMER ice sheet model and I think also the CISM model (derived from GLIMMER), the TARAH ice sheet model (Pollar ad Deconto) already embed the projection of the climate fields onto the ice sheet grid within the ice sheet code itself, and by experience with both ice sheet and climate models, I would also recommend to put OBLIMAP in the ISM code.

We once more thank for the compliments on the achievements which are realized with OBLIMAP's second release, and for summarizing the milestones from a regular user's perspective. We comment on the embedding issue in one of the next points.

Based my comments above, except minor typos or reformulations, I recommend the publication of the manuscript in its current shape. Florence Colleoni

**Major comments**

Maybe a stand-alone version, as for the first version could also be useful, not everybody uses coupled system. Or maybe this version is stand-alone but this is not clear and should be clarified with a sentence or so in manuscript.

Sure there is a stand-alone version of OBLIMAP 2.0. The stand-alone version has already been mentioned in the abstract, introduction (twice), in the caption of Fig. 1 and in the text reference to Fig. 1 in Sec. 2. However, we added:"Its stand-alone version can be installed and compiled within a couple of minutes on any platform." to the last paragraph of the conclusion, in order to emphasize once more OBLIMAP's stand-alone version and to emphasis its user friendly install.

Line 50-55: after using OBLIMAP very often, I would say that perhaps hosting OBLIMAP in the ice sheet model is the most easiest way to deal with it. OBLIMAP is a small coupler model and very simple in its use, therefore, it is easily implemented in an ISM, as for example GLIMMER does, rather than embedded in a GCM. But this is only my opinion of user. In addition the climate coupler are built to interpolate on lon-lat grids most of the time.

There might be certainly some exploration on this theme. We think that with OBLIMAP's new version OBLIMAP itself is suited for several strategies: embedding OBLIMAP in the ISM, in the GCM or in another coupler. Because OBLIMAP is subdivided in the standard components: 'Initialize', 'Run' (map or inverse map) and 'Finalize', this allows direct embedding. Note that the mapping routines pass on all fields as an argument, which makes embedding of OBLIMAP low intrusive. The strategy might depend on the specific GCM – ISM combination as well on the coupling approach: one way or two way on-line coupling. In case of a two way on-line coupling we expect that embedding the ISM in the GCM might be easier because aside of embedding OBLIMAP, the ISM has to be embedded as well in the GCM which means that the ISM needs to be recoded in an Initialize-Run-Finalize form. While vice versa the GCM has to be embedded in the ISM which probably requires recoding of the GCM in such an Initialize-Run-Finalize form which might be much more challenging due to the complexity of GCMs. Changes have been made to Sec. 3.3 and Sec. 4.3 in order to clarify our view on embedding strategies and to include the suggestion of referee 1 to add OBLIMAP's API in Sec. 3.3.

**Minor comments**

line 45: substitute "albedo changes" by "ice sheet distribution" because an ISM does not provide albedo changes, it only provides ice distribution which affect albedo within the atmosphere model. A regional atmospheric model as RACMO on the contrary provides albedo changes, but this is not an ISM.

Done

line 48: "surface mass balance" put a space between "mass" and "balance"

Done

line 52: substitute "ocean surface temperatures" by "ocean temperatures.". Most of the basal melting methods uses vertical ocean temperature and salinity distribution

(Holland and Jenkins 1999, Pollard and Deconto 2012, Martin et al., 2011).

Done

line 63: substitute by "any regional energy balance model (e.g. RACMO or MAR)"

Done

Figure 7: "surface mass balance" put a space between "mass" and "balance"

Done

Done